# GENERALIZABLE MOTION PLANNING VIA OPERATOR LEARNING

**Sharath Matada,**\* **Luke Bhan,**\* **Yuanyuan Shi, Nikolay Atanasov**
University of California, San Diego
`{smatada, lbhan, yyshi, natanasov}@ucsd.edu`

## ABSTRACT

In this work, we introduce a planning neural operator (PNO) for predicting the value function of a motion planning problem. We recast value function approximation as learning a single operator from the cost function space to the value function space, which is defined by an Eikonal partial differential equation (PDE). Therefore, our PNO model, despite being trained with a finite number of samples at coarse resolution, inherits the zero-shot super-resolution property of neural operators. We demonstrate accurate value function approximation at $16\times$ the training resolution on the MovingAI lab's 2D city dataset, compare with state-of-the-art neural value function predictors on 3D scenes from the iGibson building dataset and showcase optimal planning with 4-DOF robotic manipulators. Lastly, we investigate employing the value function output of PNO as a heuristic function to accelerate motion planning. We show theoretically that the PNO heuristic is $\epsilon$-consistent by introducing an inductive bias layer that guarantees our value functions satisfy the triangle inequality. With our heuristic, we achieve a $30\%$ decrease in nodes visited while obtaining near optimal path lengths on the MovingAI lab 2D city dataset, compared to classical planning methods ($A^*$, RRT$^*$). See project code https://github.com/ExistentialRobotics/PNO.

## 1 INTRODUCTION

Classical tools for motion planning in complex environments face performance degradation as the scale of the environment increases. To alleviate this issue, modern approaches introduce neural network models enabling computational efficacy (Lehnert et al., 2024). Recent connections have been made between motion planning and scientific machine learning in the context of solving partial differential equations (PDEs). Specifically, the motion planning problem can be formulated from an optimal control perspective leading to an Eikonal PDE whose solution yields the optimal value function. Recent work (Ni & Qureshi, 2023a) has explored solving the Eikonal equation via physics informed neural networks (PINNs) without the requirement of labeled data. However, as with classical PINNs, these approaches require retraining for each environment and, therefore, are computationally intractable for recomputing the solution quickly in dynamic environments governing the real-world.

In this work, we reformulate the solution to the Eikonal equation as an operator learning problem between function spaces. This new perspective enables the training of a single neural operator which maps an entire space of cost functions, with each cost function representing a separate environment, to a corresponding space of value functions. We design a new neural operator architecture, called Planning Neural Operator (PNO), that learns the solution operator of the Eikonal PDE and thus the value function. Due to our architecture design, PNO generalizes to new environments with different obstacle geometries without retraining. Furthermore, we capitalize on the resolution invariance property of neural operators, enabling training with coarse resolution data and deployment of the learned neural operator on test maps with $16\times$ the training data resolution as illustrated in Fig. 1. In summary, our contributions are given as follows:

(a) We formulate the motion planning problem as an optimal control problem whose value function satisfies the solution to the Eikonal PDE. We then prove to arbitrarily tight accuracy, the existence of a neural operator approximation to the Eikonal PDE solution operator.

---

\*equal contribution

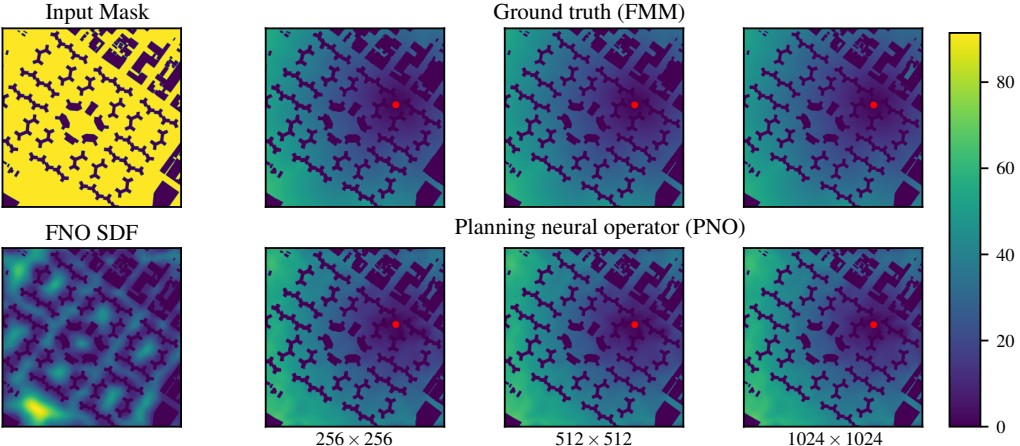

Figure 1: Example of super-resolution capabilities of PNO for motion planning on a map of NYC. The operator is trained on a dataset of resolution $64 \times 64$ and the examples shown here (resolutions $256 \times 256$, $512 \times 512$, and $1024 \times 1024$) were not seen during training. See Sec. 4 for details.

(b) We design a Planning Neural Operator (PNO) to approximate the solution operator of the Eikonal PDE and enable generalizable motion planning. By design, the architecture is (i) resolution invariant, (ii) encodes obstacle geometries into the learned weight space to enable generalization across different environments, and (iii) introduces a projection layer satisfying the triangle inequality to enable generalization across goal positions.

(c) We develop four experiments to highlight PNO's advantages. These include zero-shot super resolution, optimal path planning on 3D iGibson building environments, 4-degree of freedom (DOF) manipulators, and deployment as a heuristic function in $A^*$ reducing the expanded nodes by $33\%$ while maintaining near-optimal paths.

**Related work.** This work builds on ideas from two fields – PDEs and motion planning. Thus, we briefly review popular motion planning approaches while identifying connections between motion planning and PDEs and conclude with a brief overview of operator learning.

**Motion planning.** Motion planning techniques can classically be split into two categories – sample-based algorithms and search-based algorithms. Sampling-based motion planners generate random samples in configuration space (C-space) to construct graph structures whose edges connect to form collision-free paths. The two most widely known algorithms are probabilistic roadmaps (PRM) (Kavraki et al., 1996) and rapidly-exploring random trees (RRT) (LaValle, 2006; Karaman & Frazzoli, 2011) Since the development of these methods, there have been significant extensions using ideas such as biased sampling with heuristic functions (Gammell et al., 2015; 2014), parallelization (Sundaralingam et al., 2023), domain decomposition (Chamzas et al., 2021), and potential fields (Qureshi & Ayaz, 2015). We refer readers to Orthey et al. (2024) for a review.

In contrast with sampling-based planners, search-based planners trade speed in favor of obtaining optimal paths. The most well known approaches are Dijkstra's algorithm (Dijkstra, 1959) and $A^*$ (Hart et al., 1968; Likhachev et al., 2003), which prioritizes the search order using a heuristic function. More recent search-based planners include Cohen et al. (2010), Liu et al. (2018) which plan using motion primitives. Lastly, we briefly mention planners based on Eikonal solvers such as Sethian (1996), Janson et al. (2015), Valero-Gomez et al. (2013), Pêtrès et al. (2005). These approaches, instead of planning by exploring nodes in configuration space, aim to solve the Eikonal PDE yielding an optimal function for the resulting environment and then perform corresponding gradient descent.

Despite the success of classical planning methods, they suffer from computational limitations when scaling to higher dimensions. Thus, with the recent success of deep learning, neural motion planners (NMPs) have exploded in popularity for reducing the computational burden of motion planning. For example, many works such as value iteration networks (VIN) (Tamar et al., 2016), motion planning networks (MPN) (Qureshi et al., 2021) as well as more recent approaches such as Zeng et al. (2019),Chaplot et al. (2021) and Fishman et al. (2022) attempt to plan completely from neural networks designs. Moreover, other approaches instead augment classical algorithms. For example,

Chen et al. (2020), Zhang et al. (2022) speedup the collision checker in RRT and Ichter et al. (2019), Wang et al. (2020) learn sampling biases. See McMahon et al. (2022) for a comprehensive review. Further, some approaches use gradient information directly from either Neural Distance Fields (Ortiz et al., 2022; Liu et al., 2022) or Neural Radiance Fields (NeRF) (Adamkiewicz et al., 2022). Lastly, we mention three approaches that are most close to ours. The first, NTFields/P-NTFields (Ni & Qureshi, 2023a;b) learns a PINN for motion planning based on the Eikonal equation, but requires retraining for new environments. Meanwhile Li et al. (2022) learns implicit functions for planning but has no guarantees of obstacle avoidance and pays large startup costs for data generation.

**Operator learning.** The goal of operator learning is the design of models for approximating infinite-dimensional mappings across function spaces. Neural operators were first introduced in a seminal paper by Chen & Chen (1995) that designed the first architecture and a corresponding universal operator approximation theorem for approximating PDE solutions. This work was recently rediscovered and extended using modern neural network models under the DeepONet and FNO frameworks (Lu et al., 2021; Deng et al., 2022; Lanthaler et al., 2022; Li et al., 2021; Kovachki et al., 2023; 2021). Since, there have been a series of expansive papers addressing numerous challenges from non-uniform geometries (Liu et al., 2023; Li et al., 2023a;b; Fang et al., 2024) to architectural limitations (Seidman et al., 2022; You et al., 2022a; Gupta et al., 2021; Hao et al., 2023; Furuya et al., 2023) of the original works. Furthermore, neural operators have been employed in an eclectic range of applications spanning, but not limit to, weather forecasting (Kurth et al., 2023), material modeling (You et al., 2022b), seismic wave propagation (Yang et al., 2021), and control of PDEs (Bhan et al., 2023). Lastly, from a theoretical perspective, Lanthaler et al. (2023) unified the neural operator framework, introducing an abstract formulation of the operator learning problem and a corresponding universal approximation theorem (see Theorem 2) that encompasses a wide array of architectures, including Fourier neural operator (FNO) and DeepONet.

## 2 EIKONAL PDE FORMULATION OF THE MOTION PLANNING PROBLEM

### 2.1 MOTION PLANNING AS AN OPTIMAL CONTROL PROBLEM

Consider a continuous-time dynamical system with state $\boldsymbol{x}(t) \in \mathcal{X} \subset \mathbb{R}^n$, compact state space $\mathcal{X}$, control input $\boldsymbol{u}(t) \in \mathcal{U} \subset \mathbb{R}^k$, compact control space $\mathcal{U}$, and dynamics:

$$\dot{\boldsymbol{x}}(t) = f(\boldsymbol{x}(t), \boldsymbol{u}(t)). \tag{1}$$

Given an initial condition $\boldsymbol{x}(0) = \boldsymbol{x}_0$, we aim to design a control policy $\pi : \mathcal{X} \to \mathcal{U}$ that maintains the system state $\boldsymbol{x}(t)$ in a *safe set* $\mathcal{S} \subset \mathcal{X}$ for all $t$ and drives it to a *goal set* $\mathcal{G} \subset \mathcal{S}$. We formulate this problem through an infinite-horizon first-exit optimal control problem (OCP) as Bertsekas (2017)

$$\min_{\pi} \quad c_\tau(\boldsymbol{x}(\tau)) + \int_0^\tau c(\boldsymbol{x}(t), \pi(\boldsymbol{x}(t))) dt \,, \tag{2a}$$

$$\text{s.t.} \quad \tau = \inf\{t \in \mathbb{R}_{\geq 0} \mid \boldsymbol{x}(t) \in \mathcal{G}\}, \tag{2b}$$

$$\dot{\boldsymbol{x}}(t) = f(\boldsymbol{x}(t), \pi(\boldsymbol{x}(t))) \,, \ \boldsymbol{x}(0) = \boldsymbol{x}_0, \quad \boldsymbol{x}(t) \in \mathcal{S} \,, \ \forall t \geq 0 \,, \tag{2c}$$

where $\tau$ is the *first-exit time*, the smallest time at which the system reaches the goal region $\mathcal{G}$, $c : \mathcal{X} \times \mathcal{U} \to \mathbb{R}$ is the *stage cost function* that measures the quality of the system trajectory, and $c_\tau : \mathcal{X} \to \mathbb{R}$ is a *terminal cost function* evaluated when the system reaches the goal region. To ensure the well-posedness of (2), we enforce the following standard assumption (Liberzon, 2012). In practice, this assumption is satisfied when there is a feasible path from every start position to the goal.

**Assumption 1.** *There exists a policy $\pi : \mathcal{X} \to \mathcal{U}$ such that, for any initial condition $\boldsymbol{x}(0) \in \mathcal{S}$, the trajectories of the closed-loop system with $\boldsymbol{u}(t) = \pi(\boldsymbol{x}(t))$ in (1), reach the goal region $\mathcal{G}$ by a finite first-exit time $\tau < \infty$.*

As standard, the optimal value function $V : \mathcal{X} \to \mathbb{R}$ is the minimum cost attained in (2) given by

$$V(\boldsymbol{x}) := \min_{\pi}\{c_\tau(\boldsymbol{x}(\tau)) + \int_0^\tau c(\boldsymbol{x}(t), \pi(\boldsymbol{x}(t))) dt\}, \tag{3}$$

subject to the constraints in (2). A sufficient condition for optimality is that the function $V$ satisfies the Hamilton-Jacobi-Bellman equation:

$$0 = \min_{\boldsymbol{u} \in \mathcal{U}} \left\{ \nabla V(\boldsymbol{x})^\top f(\boldsymbol{x}, \boldsymbol{u}) + c(\boldsymbol{x}, \boldsymbol{u}) \right\}, \qquad \forall \boldsymbol{x} \in \mathcal{S} \setminus \mathcal{G}, \tag{4a}$$

$$V(\boldsymbol{x}) = c_\tau(\boldsymbol{x}), \qquad \forall \boldsymbol{x} \in \mathcal{G}, \tag{4b}$$

where $\nabla V$ denotes the gradient of $V$. As in (4a), (4b), to simplify notation, we omit the time argument for $\boldsymbol{x}(t)$ and $\pi(\boldsymbol{x}(t))$ in the remainder of the paper.

## 2.2 Eikonal equation for optimal motion planning

In this work, we capitalize on the structure of common motion planning problems to simplify the HJB equation. In practice, one commonly splits the OCP described above into two subproblems: i) a motion planning problem, which determines a reference path from the initial state $\boldsymbol{x}(0)$ to the goal region $\mathcal{G}$ without considering the dynamics constraint in (2c), and ii) a control problem, which determines a control policy for the system to track the planned reference path. We focus on the planning problem, which simplifies the system dynamics to be fully actuated, $\dot{\boldsymbol{x}}(t) = \boldsymbol{u}(t)$.

To ensure feasible paths, it is common to limit the magnitude of the control input $\boldsymbol{u}(t)$. Thus, we constrain the input to an admissible control set $\mathcal{U} = \left\{ \boldsymbol{u} \in \mathbb{R}^k \mid \|\boldsymbol{u}\| = 1 \right\}$ of unit vectors. When the input magnitude is already constrained by $\mathcal{U}$, it is not necessary to penalize it in the stage cost. Hence, we let $c(\boldsymbol{x}, \boldsymbol{u}) = c(\boldsymbol{x})$ be a control-independent cost, and let $c_\tau(\boldsymbol{x}) = c(\boldsymbol{x})$ for presentation simplicity. Thus, the optimal control problem in (2) reduces to the *optimal motion planning* problem:

$$\min_\pi \ c(\boldsymbol{x}(\tau)) + \int_0^\tau c(\boldsymbol{x}(t)) dt, \tag{5a}$$

$$\text{s.t. } \tau = \inf\{t \in \mathbb{R}_{\geq 0} \mid \boldsymbol{x}(t) \in \mathcal{G}\}, \tag{5b}$$

$$\dot{\boldsymbol{x}}(t) = \pi(\boldsymbol{x}(t)), \ \boldsymbol{x}(0) = \boldsymbol{x}_0, \tag{5c}$$

$$\boldsymbol{x}(t) \in \mathcal{S}, \quad \|\pi(\boldsymbol{x}(t))\| = 1, \quad \forall t \geq 0, \tag{5d}$$

and the associated HJB equation in (4) simplifies to:

$$0 = \min_{\boldsymbol{u} \in \mathcal{U}} \left\{ \nabla V(\boldsymbol{x})^\top \boldsymbol{u} + c(\boldsymbol{x}) \right\}, \qquad \forall \boldsymbol{x} \in \mathcal{S} \setminus \mathcal{G}, \tag{6a}$$

$$V(\boldsymbol{x}) = c(\boldsymbol{x}), \qquad \forall \boldsymbol{x} \in \mathcal{G}. \tag{6b}$$

Note that the minimization in (6a) is now linear in $\boldsymbol{u}$ with the constraint $\|\boldsymbol{u}\| = 1$. The minimizer is readily computable in closed-form $\boldsymbol{u} = -\frac{\nabla V(\boldsymbol{x})}{\|\nabla V(\boldsymbol{x})\|}$, yielding a PDE in the Eikonal class:

$$\|\nabla V(\boldsymbol{x})\| = c(\boldsymbol{x}), \qquad \forall \boldsymbol{x} \in \mathcal{S} \setminus \mathcal{G}, \tag{7a}$$

$$V(\boldsymbol{x}) = c(\boldsymbol{x}), \qquad \forall \boldsymbol{x} \in \mathcal{G}. \tag{7b}$$

## 2.3 Properties and universal approximation of the Eikonal solution operator

The solution to the Eikonal PDE in (7) can be viewed as an infinite-dimensional nonlinear operator $\Psi$ that maps cost functions $c(\boldsymbol{x})$ into corresponding value function solutions $V(\boldsymbol{x})$. However, given an arbitrary cost function $c$, there is no known analytical representation for the operator $V = \Psi(c)$. Thus, in this work, we ask if it is possible to learn an approximation to the operator $\Psi$ and, if so, how to design a resolution invariant neural network to approximate $\Psi$? To start, we answer the first question affirmatively by theoretically proving the existence of such a neural operator approximation.

To theoretically show the existence of $\hat{\Psi}$, we begin with a brief review of neural operators. A *neural operator* $\hat{\Psi}$ is a neural network architecture consisting of a lifting neural network, a kernel approximation neural network, and a projection neural network (See Appendix B for full formalization). Under such an architecture several universal approximation theorems exist of which we focus on (Lanthaler et al., 2023, Theorem 2.1) (restated as Theorem 2 in Appendix B.1) due to its abstract formulation that encompasses a variety of architectures.

Under the universal approximation theorem, there are two challenges to establishing the existence of a neural operator $\hat{\Psi}$ for approximating the Eikonal PDE solution. First, we require that the domain is a compact set of continuous functions and second we require continuity of the operator. For the first

condition, since the solution to the Eikonal PDE is not continuous, we aim to learn the continuous and unique viscosity solution (see Appendix C). Let $\mathcal{F}_c$ be the function space of costs $\mathcal{X} \to \mathbb{R}$, namely $c(\cdot) \in \mathcal{F}_c$. Likewise, let $\mathcal{F}_v$ be the function space of value functions $\mathcal{X} \to \mathbb{R}$, that is $V(\cdot) \in \mathcal{F}_v$. To ensure well-posed solutions of Problem (7), we require the following assumption.

**Assumption 2** (Continuity and uniform boundedness of cost function space). *The cost function space $\mathcal{F}_c$ is uniformly equicontinuous. Further, there exists a constant $\theta > 0$ such that*

$$\inf_{\boldsymbol{x} \in \mathcal{X}} c(\boldsymbol{x}) > 1/\theta, \quad \sup_{\boldsymbol{x} \in \mathcal{X}} c(\boldsymbol{x}) < \theta, \quad \forall c \in \mathcal{F}_c. \tag{8}$$

The set $\mathcal{F}_c$ needs to be equicontinuous to ensure the continuity of the operator $\Psi$. The uniform positivity of the cost $c$ is required to ensure the existence and uniqueness of viscosity solutions to the Eikonal PDE. In practice, this assumption can be satisfied by creating smooth boundaries around each obstacle associated with strictly positive motion cost. Lastly, by the continuity of cost functions in Assumption 2 and the Arzelá Ascoli theorem (Folland, 1999, Theorem 4.43), $\mathcal{F}_c$ is compact. From these assumptions, a viscosity solution $\Psi : \mathcal{F}_c \to \mathcal{F}_v$ to the Eikonal PDE always (7) exists (See Appendix C). We are now ready to present the existence of a neural operator approximation to $\Psi$.

**Theorem 1.** *(See Appendix A.1 for proof) Let Assumptions 1, 2 hold and consider $\Psi : \mathcal{F}_c \to \mathcal{F}_v$ as the solution to (7). Then, for any $\epsilon > 0$, there exists a neural operator $\hat{\Psi} : \mathcal{F}_c \to \mathcal{F}_v$ such that*

$$\sup_{c \in \mathcal{F}_c} \|\Psi(c) - \hat{\Psi}(c)\| \leq \epsilon. \tag{9}$$

While Theorem 1 guarantees the *existence* of a neural operator for approximating the Eikonal PDE, it does not include the construction of the neural operator $\hat{\Psi}$. This is the focus of the next section.

## 3 DESIGNING NEURAL OPERATORS FOR EIKONAL PDEs

We introduce a new neural operator architecture, which we call *Planning Neural Operator* (PNO), to approximate the solution operator $\Psi(c) = V$ of the Eikonal PDE in (7). We ensure that our architecture achieves three goals: (i) invariance to resolution differences between train and test maps, (ii) generalization across environment geometries, and (iii) generalization across goal positions.

To achieve property (i), we employ the resolution-invariant Fourier neural operator (FNO) architecture (Li et al., 2021) with two key extensions. First, to enable generalization across environments, we *hard encode* the obstacle geometries into the operator structure. Second, to generalize across goals, we design the output layer of the FNO model to ensure that the predicted value function satisfies the triangle inequality. Particularly, when the goal set is a singleton $\mathcal{G} = \{\boldsymbol{g}\}$, let $V(\boldsymbol{x}, \boldsymbol{g})$ denote the solution to (7) with goal $\boldsymbol{g}$. Then, by the principle of optimality (Bellman, 1957), $V$ satisfies:

$$V(\boldsymbol{x}, \boldsymbol{g}) \leq V(\boldsymbol{x}, \boldsymbol{y}) + V(\boldsymbol{y}, \boldsymbol{g}), \quad \forall \boldsymbol{x}, \boldsymbol{y}, \boldsymbol{g} \in \mathcal{S}. \tag{10}$$

Thus, we seek a neural operator $V(\boldsymbol{x}, \boldsymbol{g}) = (\hat{\Psi}(\boldsymbol{c}; \boldsymbol{g}))(\boldsymbol{x})$ that encodes this property of the value function. Next, we review the structure of the FNO model, which ensures property (i). Then, we present our PNO architecture that extends the FNO model to enable properties (ii) and (iii).

### 3.1 REVIEW OF FNO AND DAFNO

An FNO model consist of three components: (1) a lifting neural network that maps to high dimensional space, (2) a series of Fourier layers, and (3) a projection neural network to the target resolution. It can be viewed as a mapping $\hat{\Psi} = Q \circ L_M \circ \cdots \circ L_1 \circ R$, where $R(c(\boldsymbol{x}), \boldsymbol{x})$ is the lifting network, $L_m$ for $m = 1, \ldots, M$ are the hidden Fourier layers, and $Q$ is the projection network (See Appendix B.1 for details). The Fourier layers consist of applying a Fast Fourier transform (FFT) to the input, multiplying that transform by a learnable weight matrix and then transforming back via an Inverse FFT. Mathematically, this is formalized as $L_m(x) := \mathscr{F}^{-1}(W_{\theta_m} \cdot \mathscr{F}(x))$ where $\mathscr{F}, \mathscr{F}^{-1}$ are the Fourier and inverse Fourier transforms and $W_{\theta_m}$ is a learnable, complex valued weight matrix. The key idea is that, by learning in frequency space, the operator is resolution invariant as one can project any desired resolution to and from a pre-specified number of Fourier modes.

However, to perform a Fourier transform, FNO needs a rectangular domain and thus cannot be applied to non-uniform geometries. Recently, a new model, Domain Agnostic FNO (DAFNO) (Liu et al., 2023) addressed this issue. DAFNO circumscribes the non-uniform geometry of the solution

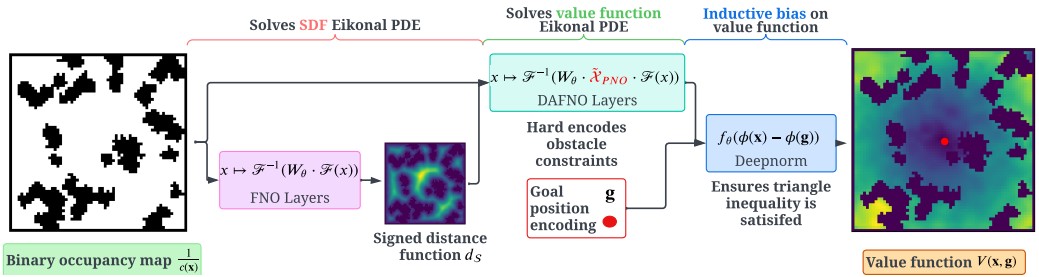

Figure 2: PNO network architecture. The input to a PNO is a binary occupancy grid, which is transformed into a sign distance function (SDF) via an independently trained FNO. This, along with the original binary map is passed to a series of modified FNO layers which hard encode the obstacles. Finally, this result, along with the goal, is then fed to a projection layer (ensuring satisfaction of the triangle inequality) obtaining the final value function prediction.

operator into a rectangle and then encodes, in the Fourier layer, where the true solution domain is active. To achieve such an encoding, DAFNO explicitly introduces an indicator matrix of the active solution geometry $\tilde{\chi}$ into the Fourier layer multiplication. Explicitly, this is expressed by the following modified layer $L_{m,\text{DAFNO}} := \mathscr{F}^{-1}(W_{\theta_m} \cdot \tilde{\chi} \cdot \mathscr{F}(x))$ where $\tilde{\chi}$ is the aforementioned indicator function (Details on the DAFNO architecture are in Appendix B.2). In PNO, we take inspiration from DAFNO by explicitly encoding the *obstacles* using a similar indicator function.

## 3.2 PNO ARCHITECTURE: GENERALIZING ACROSS ENVIRONMENTS

The PNO model maps a cost function to the corresponding value function of the motion planning problem by explicitly designing the layers $R, L_1, \ldots, L_M, Q$ to achieve goals (ii) and (iii). However, before discussing the layer design, we begin by formalizing the input to a PNO. We consider a minimum-time motion planning problem by choosing the cost function as

$$c(\boldsymbol{x}) = \begin{cases} 1 & \boldsymbol{x} \in \mathcal{S}, \\ \infty & \boldsymbol{x} \in \mathcal{X} \backslash S, \end{cases} \tag{11}$$

where we penalize the unsafe set with infinite cost. In practice, to avoid numerical instability, we use $1/c(\boldsymbol{x})$, or a binary occupancy map as the input function. We then employ a neural network for our lifting layer as in FNO, that is $R_{PNO}(1/c(\boldsymbol{x})) := W_{\theta_R}(1/c(\boldsymbol{x})) + B_{\theta_R}$ where $W_{\theta_R}$ and $B_{\theta_R}$ are learnable weight tensors. To achieve property (ii), we ensure that PNO explicitly captures the obstacle configuration of the environment. That is, we *hard encode* the obstacle locations in the Fourier layer of our architecture. To do so, taking inspiration from DAFNO, we modify the Fourier layer by multiplying the learnable weight matrix with the following smoothed indicator function

$$\tilde{\chi}_{PNO}(\boldsymbol{x}) := \tanh(\beta d_{\mathcal{S}}(\boldsymbol{x}))(1/c(\boldsymbol{x}) - 0.5) + 0.5, \tag{12}$$

where $\beta$ is a smoothing hyperparameter

$$d_{\mathcal{S}}(\boldsymbol{x}) = \begin{cases} \inf_{\boldsymbol{y} \in \partial \mathcal{S}} \|\boldsymbol{x} - \boldsymbol{y}\|, & \text{if } \boldsymbol{x} \in \mathcal{S} \\ -\inf_{\boldsymbol{y} \in \partial \mathcal{S}} \|\boldsymbol{x} - \boldsymbol{y}\|, & \text{if } \boldsymbol{x} \notin \mathcal{S}, \end{cases} \tag{13}$$

where $\partial \mathcal{S}$ is the boundary of $\mathcal{S}$. Thus, the inner layers of a PNO are given by $L_{m,PNO}(x) := \mathscr{F}^{-1}(W_{\theta_m} \cdot \tilde{\chi}_{PNO} \cdot \mathscr{F}(x))$, $m = 1, \ldots, M$. Multiplication by $\tilde{\chi}_{PNO}$ achieves two goals. First, it ensures that the indicator approximation in (12) is continuous and thus retains the guarantees of Theorem 1. Second, for each environment, PNO ignores the unsafe space in the weight matrix multiplication, which greatly improves generalization to new obstacle configurations.

In large motion planning problems, computing the SDF in the smoothing function (12) can be computationally challenging. However, note that *a SDF $d_{\mathcal{S}}(\boldsymbol{x})$ is itself a solution to an Eikonal PDE*:

$$\|\nabla d_{\mathcal{S}}(\boldsymbol{x})\| = 1, \quad \boldsymbol{x} \in \mathcal{X}, \qquad d_{\mathcal{S}}(\boldsymbol{x}) = 0, \quad \boldsymbol{x} \in \partial \mathcal{S}. \tag{14}$$

Thus, under Theorem 1, there exists a neural operator approximation to (14). As such, we introduce a second neural operator, in the form of an FNO, trained independently, that maps from a binary occupancy map to the corresponding SDF, namely $1/c(\boldsymbol{x}) \mapsto d_{\mathcal{S}}(\boldsymbol{x})$. Thus, the occupancy function

$1/c(\boldsymbol{x})$ and the SDF $d_{\mathcal{S}}(\boldsymbol{x})$ are taken as inputs to generate the smoothed indicator $\tilde{\chi}$ in (12). This is then used to modulate the kernel in the Fourier layers of PNO. Since we use an FNO to generate the SDF function, our entire architecture maintains property (i), namely resolution invariance. Lastly, inspired by PINNs for motion planning Ni & Qureshi (2023a), we introduce a PINN loss:

$$\text{Loss}(V, \hat{V}) := \|V - \hat{V}\|_{L^2} + \xi \left( \int_{x \in \mathcal{S}} (\|\nabla \hat{V}(\boldsymbol{x}, \cdot)\| - c(\boldsymbol{x}))^2 \right)^{1/2}, \tag{15}$$

where $\xi$ controls the weighting of the PINN component. Note, if $\xi \gg 1$, the PINN loss dominates resulting in any solution that satisfies the gradient (e.g. Euclidean norm) and thus must be tuned.

### 3.3 PNO ARCHITECTURE: GENERALIZING ACROSS GOAL POSITIONS

Given the lifting layer and the modified Fourier layers discussed above, we move to design the projection layer of PNO. The projection layer contains two inputs, namely the output from the last hidden layer and the goal location $\boldsymbol{g}$. This goal location is equivalent to passing the boundary condition (7b) for the solution operator $\Psi$ into our network. To enable goal generalization (iii), we leverage the fact that the optimal value function must satisfy the triangle inequality. Thus, we design our output with an inductively biased Deepnorm layer $Q(\cdot, \cdot)$ (Pitis et al., 2020) given by:

$$Q_{PNO}(\phi, \boldsymbol{x}, \boldsymbol{g}) = f_{\theta_Q}(\phi(\boldsymbol{x}) - \phi(\boldsymbol{g})), \tag{16}$$

where $\phi = L_{M,PNO} \circ \cdots \circ L_{1,PNO} \circ R_{PNO}$, and thus $\phi(\boldsymbol{x}) - \phi(\boldsymbol{g})$ is the subtraction of the feature vectors between $\boldsymbol{x}$ and the goal $\boldsymbol{g}$. $f_{\theta_Q}$ consists of regular neural network layers with a *non-negative* activation function (e.g., ReLU) and a *positive* weight matrix $W^+$. This ensures that our value function satisfies the triangle inequality with respect to $\boldsymbol{g}$ (Pitis et al., 2020) and enables generalization across different goal positions (Section 4). Lastly, we summarize the PNO architecture in Fig. 2, starting from a binary occupancy input $1/c(\boldsymbol{x})$ and returning a value function as the output.

## 4 LEARNING MOTION PLANNING VALUE FUNCTIONS

To validate the efficacy of our PNO architecture, we design four experiments. First, we test our method on grid-world environments comparing PNO against learning-based motion planners. However, given the simplicity of the small grid world dataset, the environments lack the complexity of real-world tasks. Thus, we introduce three *real-world* experiments. The first experiment highlights the super-resolution property of PNO by training on small synthetic maps and evaluating on large real-world maps from the Moving AI 2D city dataset (Sturtevant, 2012). In remaining experiments, we showcase the scalability of our approach in 3D iGibson environments (Shen et al., 2021) and 4DOF manipulators.

We compare our method against the FMM (Sethian, 1996) which solves the Eikonal PDE numerically, state-of-the-art neural motion planners: VIN (Tamar et al., 2016), NTFields (Ni & Qureshi, 2023a), P-NTFields Ni & Qureshi (2023b), IEF2D (Li et al., 2022), and two operator learning architectures: FNO (Li et al., 2021), DAFNO (Liu et al., 2023) (Baselines details in Appendix D).

**Grid-world environments.** In the first experiment, we consider a small Grid-World dataset as in Tamar et al. (2016), consisting of 5k training maps and 1k testing maps. We compare with VIN and IEF2D. For planning, we perform gradient descent on the test-map value function predictions from VIN, IEF2D, and PNO. In Table 1, we present the computation time needed for value function prediction and the success rate of reaching the goal. We

Table 1: Comparison of planning on learned value functions on the Grid World dataset at various sizes.

| | Avg. success rate ↑ | | | Avg. computation time (ms) ↓ | | |
|---|---|---|---|---|---|---|
| | $8^2$ | $16^2$ | $28^2$ | $8^2$ | $16^2$ | $28^2$ |
| VIN | 99.6 | **99.3** | 96.7 | 3.9 | 22.7 | 82.1 |
| IEF2D | 99.7 | 98.3 | 97.0 | 13.3 | 14.8 | 20.4 |
| PNO (ours) | **99.9** | 97.3 | **99.3** | **2.6** | **4.7** | **6.4** |

can see that all baselines perform quite well, with PNO outperforming both methods on the larger $28 \times 28$ sized mazes while achieving an improved computational time. In Appendix D, we give an example of the learned value function and corresponding map for the PNO. The superior performance of PNO on the Grid World dataset is primarily due to the fact PNO satisfies the triangle inequality and guarantees obstacle avoidance where as both VIN and IEF2D lack such properties.

**Ablation study: Moving AI 2D city maps.** To highlight the advantage of reformulating motion planning as an operator learning problem in continuous function space, we demonstrate that our

PNO architecture can be trained with a *coarse* resolution and deployed at a *finer* super-resolution on the Moving AI 2D real-world city maps (see Fig. 1). For training, we use a synthetic map dataset generated at $64 \times 64$ resolution (See Appendix D.3). We then evaluate the super-resolution performance of our method in comparison with two operator learning architectures in Table 2. These results constitute an ablation study of the obstacle encoding and Deepnorm components of our PNO model (see Fig. 2). In particular, DAFNO and PNO outperform FNO on the synthetic and real-world datasets, highlighting the importance of explicitly encoding the obstacle geometry into the network structure. Furthermore, we see the effect of the Deepnorm layer as PNO generalizes better compared to DAFNO, particularly in the unseen real-world city dataset achieving a $50\%$ reduction in $L_2$ error. In the third column of Table 2, we see the computational speedup at scale of the PNO architecture. In particular, the SDF generation significantly slows DAFNO and we achieve almost $10\times$ speedup over the fast marching method (FMM). Lastly, we provide an example of navigating New York City in Fig. 1 where we showcase that the FNO is able to learn the SDF accurately (relative $L_2$ test error of $0.064$) and that the corresponding value function is accurately computed at all three super-resolution scales.

Table 2: Average $L_2$ value function error of FNO, DAFNO, and PNO versus FMM.

| Map Size | Avg. relative L2 error ↓ | | | | | | | | Avg. computation time ↓ | | | |
|---|---|---|---|---|---|---|---|---|---|---|---|---|
| | $64^2$ | $256^2$ | $512^2$ | $1024^2$ | $64^2$ | $256^2$ | $512^2$ | $1024^2$ | $64^2$ | $256^2$ | $512^2$ | $1024^2$ |
| | Synthetic obstacle dataset (100 maps, in-distribution) | | | | MovingAI real-world city dataset (90 maps, out-of-distribution) | | | | Synthetic + real-world city (1000 maps, ms) | | | |
| **FNO** (PNO w/o Deepnorm and obstacle encoding) | 0.1996 | 0.5771 | 0.6214 | 0.6405 | — | 0.7188 | 0.7519 | 0.7692 | 1.62 | **1.64** | **1.66** | **2.28** |
| **DAFNO** (PNO w/o Deepnorm layer) | 0.0985 | 0.3868 | 0.4060 | 0.4120 | — | 0.4090 | 0.4259 | 0.4315 | 3.10 | 4.97 | 11.44 | 49.82 |
| **PNO** (ours) | 0.1136 | 0.1197 | 0.1190 | 0.1194 | — | 0.1748 | 0.1885 | 0.2034 | 5.57 | 5.31 | 6.33 | 8.31 |
| **PNO w/ PINN loss** (ours) | **0.0698** | **0.0865** | **0.0869** | **0.0872** | — | **0.1675** | **0.1761** | **0.1842** | 5.57 | 5.31 | 6.33 | 8.31 |
| **FMM** (numerical solver) | — | — | — | — | — | — | — | — | **0.34** | 5.96 | 25.66 | 104.64 |

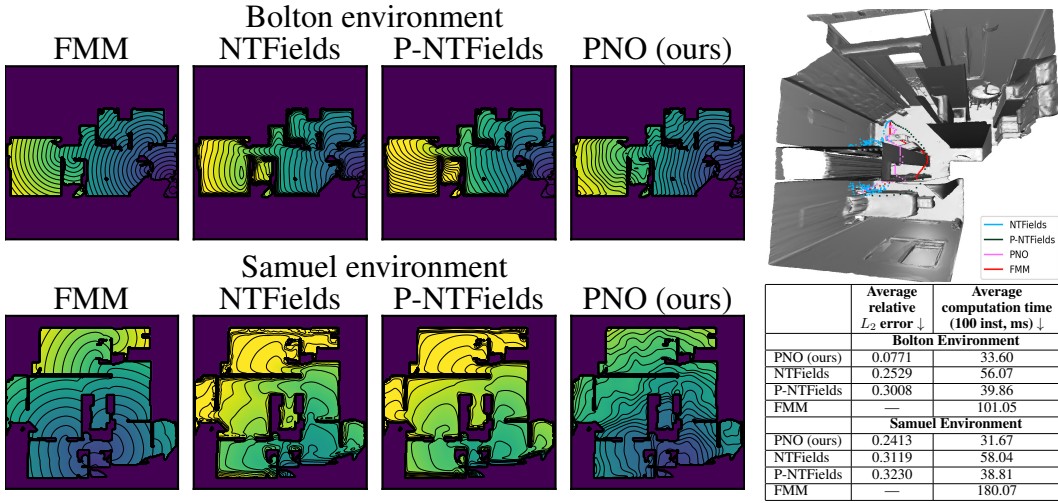

Figure 3: (left) Comparison of 3D value function approximations for two iGibson environments (best $L_2$ PNO, sliced at $z = 0$). (right) On top, we show example paths in the Bolton environment. On the bottom table, we present a aggregate quantitative comparison. See Appendix D.4 for more examples.

**3D planning in iGibson environments.** In this experiment, we consider planning in 3D real-world iGibson environments (details in Appendix D.4). For comparison, we train NTFields and P-NTFields models on the Bolton and Samuel environments which takes approximately 3 hours to train *per environment* (NVIDIA $A$100 GPU). To validate the generalization of our PNO, we ensure that our training dataset contains no instances of the Samuel environment, but does contain instances of the Bolton environment with different start goal positions than during testing. Generating our dataset takes approximately 1 hour and the model training takes 4 hours (NVIDIA $A$100 GPU).

Fig. 3 shows slices of the predicted value functions, where all four methods capture the general structure. On the top right, we showcase an example of planning. NTFields, due to local minima, is unable to guarantee valid paths whereas P-NTFields, PNO and FMM reach the goal successfully. In table of Fig. 3, we see that, quantitatively, our method performs well at predicting the exact value

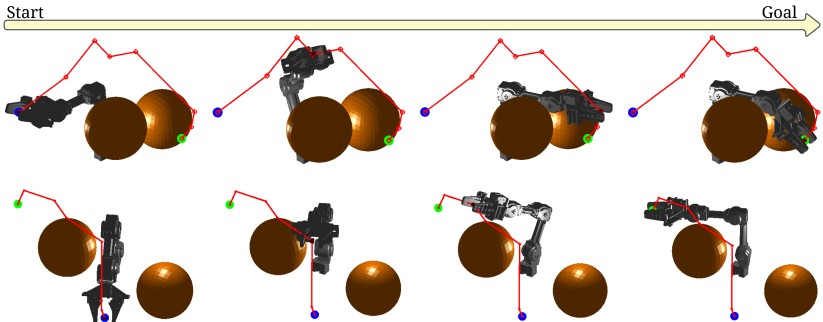

Figure 4: Planning with PNO generated value functions for a 4-DOF manipulator around two obstacles. The top and bottom rows show two separate examples with end-effector trajectory snapshots demonstrating motion from a start (blue dot) to a goal configuration (green dot).

function of FMM. Furthermore, P-NTFields does not accurately capture the exact magnitude of the FMM value function, but generates smooth paths due to effectively capturing the overall shape and thus the gradient direction. Finally, all the neural network approaches perform faster then the numerical solver achieving speedups of $2\times$ (NTFields) and $3\times$ (PNO, P-NTFields) over FMM.

**Planning with 4 DOF Manipulator.** We further highlight the efficacy of the PNO design by planning with a 4-DOF robotic manipulator. In particular, the PNO architecture learns a mapping from a 4 dimensional C-space representation of the binary occupancy grid to the corresponding value function. We learn on a grid of size $17^4$ generating 40 maps with 10 randomized goal positions for training and 10 maps with 10 randomized goal positions for testing. The PNO achieves a $0.027$ $L_2$ relative training error and a $0.041$ $L_2$ relative testing error (Details in Appendix D.5.) We show two examples of corresponding paths computed with classical gradient descent in Fig. 4. Furthermore, we showcase slices of the value function in Fig. 10 in Appendix D.5.

## 5 LEARNED OPERATOR VALUE FUNCTIONS AS NEURAL HEURISTICS

Although one can plan on the learned value functions directly, they may exhibit local minima due to approximation error. To address this challenge, we capitalize on the rich set of classical planning algorithms by employing our learned value function as a neural heuristic. Typically, motion planners such as $A^*$ use the Euclidean norm as a heuristic as it is both fast to compute and guarantees an optimal path. However, the Euclidean norm ignores the geometry of the obstacles and thus regions surrounding obstacles can require extensive exploration before an optimal path is found. Accordingly, we propose using the learned value functions from PNO as an alternative heuristic to the Euclidean norm. We prove that our heuristic is $\epsilon-$consistent and showcase the advantage numerically, achieving a $33\%$ decrease in the number of explored nodes compared to the Euclidean norm.

We begin by showing that PNO value function is an $\epsilon$-consistent heuristic, which is a sufficient property for the $A^*$ algorithm to compute an $\epsilon$-optimal path (Likhachev et al., 2003).

**Definition 1.** *Let $V(\boldsymbol{x}, \boldsymbol{g})$ be the value function with goal position $\boldsymbol{g}$. A heuristic function $h(\boldsymbol{x})$ : $\mathcal{X} \to \mathbb{R}$ is said to be **admissible** if $h(\boldsymbol{x}) \leq V(\boldsymbol{x}, \boldsymbol{g})$ for any $\boldsymbol{x} \in \mathcal{X}$. It is said to be **consistent** if $h(\boldsymbol{x}) \leq V(\boldsymbol{x}, \boldsymbol{y}) + h(\boldsymbol{y})$. Furthermore, for any $\epsilon > 1$, a heuristic is **$\epsilon$-consistent** if $h(\boldsymbol{x}) \leq \epsilon V(\boldsymbol{x}, \boldsymbol{y}) + h(\boldsymbol{y})$.*

**Lemma 1** ($\epsilon$-consistency of neural heuristic). *Let Assumption 1 hold and let $\mathcal{F}_c$ be the cost function space satisfying Assumption 2. Let $\epsilon_{NO}$ be the neural operator approximation error as in (9). Then, $\hat{V}(\boldsymbol{x})$, generated from the neural operator is an $\epsilon$-consistent heuristic with:*

$$\epsilon = \max_{\{\boldsymbol{x}, \boldsymbol{y} \in \mathcal{S} \,|\, \boldsymbol{x} \neq \boldsymbol{y}\}} 1 + 2\epsilon_{NO}/V(\boldsymbol{x}, \boldsymbol{y}). \tag{17}$$

*Further, if one is interested in a neural operator satisfying a specific $\epsilon-$consistency for $\epsilon > 1$, then the neural operator must have error no worse than:*

$$\epsilon_{NO} \leq \min_{\{\boldsymbol{x}, \boldsymbol{y} \in \mathcal{S} \,|\, \boldsymbol{x} \neq \boldsymbol{y}\}} (\epsilon - 1)/V(\boldsymbol{x}, \boldsymbol{y}). \tag{18}$$

In continuous space, it is likely that $\epsilon \to \infty$ in Lemma 1. However, motion planning algorithms, such as $A^*$, discretize the space yielding $\epsilon < \infty$. Further, in Lemma 1, it is impossible to identify the

true approximation error $\epsilon_{NO}$. Thus, we introduce a second, practical improvement. Our key idea is to "erode" the obstacles (as in Fig. 5) with the intuition the neural operator applied to the eroded environment is more likely to be an under-approximation to the true value function. This encourages *admissibility* which can improve the paths generated under the heuristic (Pearl, 1984). To conduct the erosion, we remove the outermost layer from each obstacle (Serra, 1983) and repeat this operation several times depending on the environment size. Formally, this can be expressed as increasing the safe space as $\tilde{\mathcal{S}} \supset \mathcal{S}$, where $\tilde{\mathcal{S}}$ is the safe set after erosion and then computing the cost as in (11) with $\tilde{\mathcal{S}}$. The following result guarantees that our eroded PNO heuristic does not harm the $\epsilon$-consistency.

**Lemma 2** (Eroded heuristic is "more" consistent). *Let Assumptions 1, 2 hold. For any $\epsilon_{NO} > 0$, let $\hat{\Psi}(c), \hat{\Psi}(\tilde{c})$ be the learned solutions to (7) satisfying Theorem 1 with $\epsilon_{NO}$ according to costs $c, \tilde{c}$ defined in (11) with $\mathcal{S}, \tilde{\mathcal{S}}$ respectively. Let $\epsilon$ and $\tilde{\epsilon}$ be the $\epsilon-$consistent value functions generated by operators $\hat{\Psi}(c)$ and $\hat{\Psi}(\tilde{c})$ with $\epsilon_{NO}$ error as in Lemma 1. Then, $\epsilon \geq \tilde{\epsilon}$.*

To ensure we do not significantly underestimate the value function, we combine our heuristic with the Euclidean norm via $h(\boldsymbol{x}) = \max\{\|\boldsymbol{x} - \boldsymbol{g}\|, \hat{\Psi}(\tilde{c}(\boldsymbol{x}))\}$ while preserving $\epsilon$-consistency. We evaluate the impact of our PNO heuristic on the Moving AI lab city maps. For planning, we use $A^*$ with our PNO estimated value function as the heuristic. In Table 3, we see that the PNO heuristic with erosion achieves near optimal paths while expanding 33% fewer nodes compared to the Euclidean norm heuristic. Furthermore, note that without erosion, the PNO heuristic generates sub-optimal paths (Ablation study in Appendix E). Lastly, in Fig. 5, we highlight a single example of planning where the PNO heuristic yields an optimal path while expanding fewer nodes.

Table 3: Comparison of heuristics for $A^*$ against classical RRT and RRT$^*$ over 2D maps. The number of eroded layers is 12, 14, and 18 for $256^2$, $512^2$, and $1024^2$, respectively.

| | Avg. path length ↓ | | | $\epsilon$ suboptimality estimate ↓ | | | Avg. number of nodes expanded ↓ | | | Avg. computational time (50 inst., s) ↓ | | |
|---|---|---|---|---|---|---|---|---|---|---|---|---|
| Map size (2D) | $256^2$ | $512^2$ | $1024^2$ | $256^2$ | $512^2$ | $1024^2$ | $256^2$ | $512^2$ | $1024^2$ | $256^2$ | $512^2$ | $1024^2$ |
| A* - Euclidean norm | **144.05** | **311.21** | **605.14** | **1.000** | **1.000** | **1.000** | 3310 | 14850 | 59965 | 0.186 | 0.868 | 3.606 |
| A* - PNO (ours) w Erosion | 144.05 | 311.72 | 608.54 | **1.000** | **1.000** | 1.006 | 2024 | 9869 | 41394 | 0.136 | 0.684 | 2.759 |
| A* - PNO (ours) w/o Erosion | 144.88 | 315.25 | 614.40 | 1.005 | 1.013 | 1.015 | **1757** | **8882** | **39438** | 0.121 | 0.587 | 2.703 |
| RRT | 191.98 | 418.61 | 800.31 | 1.333 | 1.345 | 1.322 | - | - | - | **0.014** | **0.045** | **0.066** |
| RRT* | 177.29 | 403.56 | 783.83 | 1.231 | 1.297 | 1.295 | - | - | - | 0.120 | 0.600 | 2.71 |

Euclidean norm      PNO value w/o erosion      PNO value w erosion

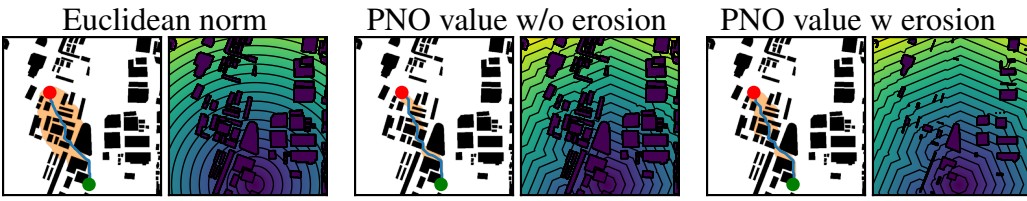

Figure 5: Path planning using various heuristics on the Moving AI Shanghai city map. Each example on the left showcases the $A^*$ planning under the corresponding heuristic given on the right. The nodes expanded are in orange, the start in red, the goal in green, and the path in blue.

## 6 CONCLUSIONS, LIMITATIONS, AND FUTURE WORK

We reformulated the motion planning problem as an Eikonal PDE and focused on learning its solution operator. To learn the operator, we developed the Planning Neural Operator (PNO) which is (i) resolution invariant, (ii) does not require retraining in new environments, and (iii) generalizes across goal positions. We evaluated our architecture on the $2D$ MovingAI city dataset and the $3D$ iGibson building dataset, showcasing super-resolution value function prediction while achieving speedups of $10\times$ over a numerical PDE solver. Lastly, we proved that our value function predictions can be used as $\epsilon$-consistent heuristics for motion planning algorithms, and demonstrated a 33% decrease in expanded nodes in the $A^*$ algorithm compared to planning with a Euclidean norm heuristic.

In the future, we aim to extend PNO with dynamic heuristic updates leveraging its fast planning inference. To address sub-optimal paths, one approach is to invoke $ARA^*$ to repair inconsistent nodes where the second iteration uses a 1-consistent heuristic (e.g. Euclidean norm). This would capitalize on PNO's efficiency while ensuring optimality. Lastly, we considered cost functions with uniform step penalties. An exciting extension would be training PNO in environments with non-uniform costs.

ACKNOWLEDGMENTS

We greatly thank Jeff Calder's (jwcalder@umn.edu) for his assistance in the proof of Theorem 1. We gratefully acknowledge support from Department of Energy (DOE) grant DE-SC0024386, Office of Naval Research ONR N00014-23-1-2353, National Science Foundation (NSF) 2120019 (CHASE-CI) and DOE DE-SC0025495. We also thank James Sorge for his assistance in various experiments. Sharath Matada thanks his family for supporting his research work throughout the Summer of 2024.

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

# APPENDIX

## A    PROOFS OF LEMMAS AND THEOREMS

### A.1    PROOF OF THEOREM 1

*Proof.* We show $\Psi$ is a continuous operator in $L^\infty$ and then invoke Theorem 2 noting that $\mathcal{F}_c$ is compact by Assumptions 2 and the Arzelá Ascoli theorem. Since, $\Psi$ is the viscosity solution, it

satisfies the comparison principle (Katzourakis, 2014, Theorem 1). Thus, let $c_1, c_2 \in \mathcal{F}_c$ such that $c_1 \leq c_2$. Then, via the comparison principle, $\Psi(c_1) \leq \Psi(c_2)$. Furthermore, by the Archimedean property, we can find a real number $\lambda > 0$ such that $\lambda c_1 \geq c_2$. A simple choice for such a $\lambda$ is $\lambda = \sup_{\boldsymbol{x} \in \mathcal{X}} (\frac{c_2}{c_1})(\boldsymbol{x})$. Then we have that $\lambda \Psi(c_1) \geq \Psi(c_2)$ again by the comparison principle. Finally, using substitution, Cauchy Schwartz, and then substitution again, we obtain

$$
\begin{aligned}
\|\Psi(c_2) - \Psi(c_1)\|_{L_\infty} &\leq \|(\lambda - 1)\Psi(c_1)\|_{L^\infty} \\
&\leq \left( \sup_{\boldsymbol{x} \in \mathcal{X}} \left( \frac{c_2 - c_1}{c_1} \right)(\boldsymbol{x}) \right) \|\Psi(c_1)\|_{L^\infty} \\
&\leq \theta_v \theta_c \|c_2 - c_1\|_{L^\infty} ,
\end{aligned}
\tag{19}
$$

where $\theta_v$ is the diameter of the value function space $\mathcal{F}_v$. Thus $\Psi$, $\mathcal{F}_c$ satisfy the requirements of Theorem 2 completing the result. □

## A.2 PROOF OF LEMMA 1

*Proof.* Let $h^*(\boldsymbol{x}) = \Psi(c)(\boldsymbol{x})$ be the optimal value function. The optimal value function is consistent, namely for any $\boldsymbol{x}, \boldsymbol{y} \in \mathcal{X}$, we have $h^*(\boldsymbol{x}) - h^*(\boldsymbol{y}) \leq V^*(\boldsymbol{x}, \boldsymbol{y})$. Now, for any $\epsilon_{NO}$, there exists some $\hat{\Psi}(c)$ satisfying Theorem 1 such that $\|\Psi(c) - \hat{\Psi}(c)\| < \epsilon_{NO}$ for any $\boldsymbol{x}, \boldsymbol{y} \in \mathcal{X}$. Then, we have

$$
\begin{aligned}
h(\boldsymbol{x}) - h(\boldsymbol{y}) &\leq h^*(\boldsymbol{x}) - h^*(\boldsymbol{y}) + 2\epsilon_{NO} \\
&\leq V^*(\boldsymbol{x}, \boldsymbol{y}) + 2\epsilon_{NO} .
\end{aligned}
\tag{20}
$$

To ensure, that (20) is $\leq \epsilon V^*(\boldsymbol{x}, \boldsymbol{y})$ for every $\boldsymbol{x}, \boldsymbol{y}$, we can choose $\epsilon_{NO} \leq \min_{\{\boldsymbol{x}, \boldsymbol{y} \in \mathcal{X} | \boldsymbol{x} \neq \boldsymbol{y}\}} \frac{(\epsilon - 1)V^*(\boldsymbol{x}, \boldsymbol{y})}{2}$. Likewise, the smallest $\epsilon$ achieved is $\epsilon \geq \max_{\{\boldsymbol{x}, \boldsymbol{y} \in \mathcal{X} | \boldsymbol{x} \neq \boldsymbol{y}\}} 1 + \frac{2\epsilon_{NO}}{V^*(\boldsymbol{x}, \boldsymbol{y})}$. □

## A.3 PROOF OF LEMMA 2

*Proof.* Let $V^*(\boldsymbol{x}, \boldsymbol{y})$ and $\tilde{V}^*(\boldsymbol{x}, \boldsymbol{y})$ be the optimal value functions for $c, \tilde{c}$ respectively. Then, $V^* \geq \tilde{V}^*$ by definition of $c, \tilde{c}$. Let $\epsilon(c), \epsilon(\tilde{c})$ be the minimum $\epsilon_2$ satisfying Lemma 1, (20) for $c, \tilde{c}$ respectively. Explicitly

$$
\epsilon(c) = \max_{\{\boldsymbol{x}, \boldsymbol{y} \in \mathcal{X} | \boldsymbol{x} \neq \boldsymbol{y}\}} \frac{V^*(x, y) + 2\epsilon_{NO}}{V^*(x, y)} ,
\tag{21}
$$

$$
\epsilon(\tilde{c}) = \max_{\{\boldsymbol{x}, \boldsymbol{y} \in \mathcal{X} | \boldsymbol{x} \neq \boldsymbol{y}\}} \frac{\tilde{V}^*(x, y) + 2\epsilon_{NO}}{V^*(x, y)} .
\tag{22}
$$

Now, noting that $\tilde{V}^* \leq V^*$ everywhere yields $\epsilon(\tilde{c}) \leq \epsilon(c)$. □

## B NEURAL OPERATORS

### B.1 NONLOCAL NEURAL OPERATORS

Formally, we provide a review of the nonlocal neural operator (NNO) as in Lanthaler et al. (2023) under the architecture of the general neural operator first introduced in Kovachki et al. (2023). Such a framework is useful as it encompasses almost all neural operator architectures including the well known DeepONet Lu et al. (2021) and FNO Li et al. (2021). Let $\mathcal{X} \subset \mathbb{R}^n$ be a bounded domain and define the following function spaces consisting of continuous functions $\mathcal{F}_c \subset C^0(\mathcal{X}; \mathbb{R})$, $\mathcal{F}_v \subset C^0(\mathcal{X}; \mathbb{R})$. Then, a NNO is defined as a mapping $\hat{\Psi} : \mathcal{F}_c(\mathcal{X} : \mathbb{R}) \to \mathcal{F}_v(\mathcal{X}; \mathbb{R})$ which can be written in the compositional form $\hat{\Psi} = \mathcal{Q} \circ \mathcal{L}_L \circ \cdots \circ \mathcal{L}_1 \circ \mathcal{R}$ consisting of a lifting layer $\mathcal{R}$, hidden layers $\mathcal{L}_l, l = 1, ..., L$, and a projection layer $\mathcal{Q}$. Given a channel dimension $d_c$, the lifting layer $\mathcal{R}$ is given by

$$
\mathcal{R} : \mathcal{F}_c(\mathcal{X}; \mathbb{R}) \to \mathcal{F}_s(\mathcal{X}; \mathbb{R}^{d_c}), \quad c(\boldsymbol{x}) \mapsto R(c(\boldsymbol{x}), \boldsymbol{x}) ,
\tag{23}
$$

where $\mathcal{F}_s(\mathcal{X}; \mathbb{R}^{d_c})$ is a Banach space for the hidden layers and $R : \mathbb{R} \times \mathcal{X} \to \mathbb{R}^{d_c}$ is a learnable neural network acting between finite dimensional Euclidean spaces. For $l = 1, ..., L$ each hidden

layer $\mathcal{L}_l$ is of the form

$$(\mathcal{L}_l v)(\boldsymbol{x}) := \sigma\left(W_l v(\boldsymbol{x}) + b_l + (\mathcal{K}_l v)(\boldsymbol{x})\right) \tag{24}$$

where weights $W_l \in \mathbb{R}^{d_c \times d_c}$ and biases $b_l \in \mathbb{R}^{d_c}$ are learnable parameters, $\sigma : \mathbb{R} \to \mathbb{R}$ is a smooth, infinitely differentiable activation function that acts component wise on inputs and $\mathcal{K}_l$ is the nonlocal operator given by

$$(\mathcal{K}_l v)(\boldsymbol{x}) = \int_{\mathcal{X}} K_l(\boldsymbol{x}, \boldsymbol{y}) v(\boldsymbol{y}) dy \tag{25}$$

where $K_l(\boldsymbol{x}, \boldsymbol{y})$ is the kernel containing learnable parameters given in various forms. For example, in the FNO architecture, $K_l(\boldsymbol{x}, \boldsymbol{y}) = K_l(\boldsymbol{x} - \boldsymbol{y})$, $K_l(\boldsymbol{x}) = \sum_{|k| \leq k_{\max}} \hat{P}_{l,k} e^{ik\boldsymbol{x}}$ is a trigonometric polynomial (Fourier) approximation with $k_{\max}$ nodes and $\hat{P}_{l,k}$ is a matrix of complex, learnable parameters $\hat{P}_{l,k} \in \mathbb{C}^{d_c \times d_c}$. Note that (24) is almost a traditional feed-forward neural network except for the kernel term (25), that is nonlocal - it depends on points over the entire domain rather then just $\boldsymbol{x}$. Lastly, the projection layer $\mathcal{Q}$ is defined as

$$\mathcal{Q} : \mathcal{F}_s(\mathcal{X}; \mathbb{R}^{d_c}) \to \mathcal{F}_v(\mathcal{X}; \mathbb{R}), \quad s(\boldsymbol{x}) \mapsto Q(s(\boldsymbol{x}), \boldsymbol{y}), \tag{26}$$

where $Q$ is a finite dimensional neural network from $\mathbb{R}^{d_c} \times \mathcal{X} \to \mathbb{R}$ yielding the final value of the operator $(\hat{\Psi} c)$ $(c \in \mathcal{F}_c)$ at the point $\boldsymbol{x} \in \mathcal{X}$.

**Theorem 2** (Neural operator approximation theorem Lanthaler et al. (2023, Theorem 2.1)). *Let $\mathcal{X} \subset \mathbb{R}^n$ be a bounded domain with Lipschitz boundary and $\overline{\mathcal{X}}$ its respective closure. Let $\Psi : C^0(\overline{\mathcal{X}}; \mathbb{R}) \to C^0(\overline{\mathcal{X}}; \mathbb{R})$ be a continuous operator, where $C^0(\overline{\mathcal{X}}; \mathbb{R})$ is the set of continuous functions $\overline{\mathcal{X}} \to \mathbb{R}$. Then for any $\epsilon > 0$ and some compact set $\mathcal{K} \subset C^0(\overline{\mathcal{X}}; \mathbb{R})$, there exists a nonlocal neural operator $\hat{\Psi} : \mathcal{K} \subset C^0(\overline{\mathcal{X}}; \mathbb{R}) \to C^0(\overline{\mathcal{X}}; \mathbb{R})$ such that*

$$\sup_{c \in \mathcal{K}} \|\Psi(c) - \hat{\Psi}(c)\|_\infty \leq \epsilon. \tag{27}$$

## B.2 Domain-Agnostic Fourier Neural Operators

Domain-Agnostic Fourier Neural Operators (DAFNO), first introduced in Liu et al. (2023), augment the FNO with a mask such that FNO's can be applied on non rectangular geometries despite the requirement of Fourier transforms to operate on periodic domains. For the motion planning problem, as above, consider partitioning $\mathcal{X}$ into two domains $\mathcal{S}$ for the safeset and $\mathcal{X} \backslash \mathcal{S}$ as the unsafe set. Furthermore, if $\mathcal{X}$ is not a box, we consider the smallest rectangle $\mathcal{T}$ that contains $\mathcal{X}$ (see Liu et al. (2023, Fig. 1)) and add all the padded points in $\mathbb{T} \backslash \mathcal{X}$ to the unsafe set. Then, consider the following two characteristic functions that encode the geometry

$$\chi(\boldsymbol{x}) := \begin{cases} 1 & \boldsymbol{x} \in \mathcal{S} \\ 0 & \boldsymbol{x} \in \mathcal{X} \backslash \mathcal{S} \end{cases}, \tag{28a}$$

$$\tilde{\chi}(\boldsymbol{x}) := \tanh(\beta d_{\mathcal{S}}(\boldsymbol{x}))(\chi(\boldsymbol{x}) - 0.5) + 0.5, \tag{28b}$$

where $\beta \in \mathbb{R}$ is a chosen hyper parameter parameter and $d_{\mathcal{S}}$ is the sign distance function (SDF) given by

$$d_{\mathcal{S}}(\boldsymbol{x}) = \begin{cases} \inf_{y \in \partial \mathcal{X}_{\mathcal{S}}} \|\boldsymbol{x} - \boldsymbol{y}\| & \text{if } \boldsymbol{x} \in \mathcal{S} \\ -\inf_{y \in \partial \mathcal{X}_{\mathcal{S}}} \|\boldsymbol{x} - \boldsymbol{y}\| & \text{if } \boldsymbol{x} \notin \mathcal{X}_{\mathcal{S}} \end{cases}. \tag{29}$$

The idea is that $\chi$ masks the geometry, setting the domain to $0$ where obstacles are and $\tilde{\chi}$ is a smoothed version to ensure that the encoded geometry is continuous satisfying Theorem 2. Then, the DAFNO architecture uses the following instantiation of the kernel in (25) as

$$(\mathcal{K}_l v)(\boldsymbol{x}) = \int_{\mathbb{T}} \tilde{\chi}(\boldsymbol{x}) \tilde{\chi}(\boldsymbol{y}) K_l(\boldsymbol{x} - \boldsymbol{y})(v(\boldsymbol{y}) - v(\boldsymbol{x})) dy, \tag{30}$$

where $K_l(\boldsymbol{x} - \boldsymbol{y})$ is defined as the trigonometric polynomials as in the original FNO architecture. Lastly, we mention the subtraction of $v(\boldsymbol{x}) - v(\boldsymbol{y})$ in (30), was introduced in You et al. (2022a) and has shown superior performance over FNOs.

## C Review of Viscosity Solutions for PDEs

**Definition 2** (Viscosity solution (Katzourakis, 2014)). *A bounded, uniformly continuous function $V : \mathcal{X} \to \mathbb{R}$ is called a **viscosity solution** of the Eikonal initial-value Problem (7) provided*

 i. $V(\boldsymbol{x}) = c(\boldsymbol{x})$ when $x \in \mathcal{G}$

 ii. *Given any function $v \in C^1(\mathcal{X})$, the following two hold*

  a. *If $V - v$ has a local maximum at the point $\boldsymbol{x} \in \mathcal{X}$, then,*

$$\|\nabla v(\boldsymbol{x})\| - c(\boldsymbol{x}) \leq 0. \tag{31}$$

  b. *If $V - v$ has a local minimum at the point $\boldsymbol{x} \in \mathcal{X}$, then,*

$$\|\nabla v(\boldsymbol{x})\| - c(\boldsymbol{x}) \geq 0. \tag{32}$$

We briefly mention that the existence of viscosity solutions can be shown in two ways - namely Perron's method via the maximum principle or via the vanishing viscosity method. We briefly detail the latter result and refer the reader to Evans (2010) for more formal analysis.

**Lemma 3.** *(Vanishing viscosity method Evans (2010)) Following the definition above, let $\Psi$ be the viscosity solution to the given problem (7). Then, let $\Psi_\epsilon$ be the classical solution to the following viscous regularized Eikonal problem*

$$\|\nabla(\Psi_\epsilon c)(\boldsymbol{x})\| + \epsilon_v \Delta(\Psi_\epsilon c)(\boldsymbol{x}) = c(\boldsymbol{x}), \quad \forall c \in \mathcal{F}_c, \boldsymbol{x} \in \mathcal{X}, \tag{33a}$$

$$\Psi(\boldsymbol{g}) = c(\boldsymbol{g}), \qquad \forall \boldsymbol{g} \in \mathcal{G}. \tag{33b}$$

*Then, $\Psi_\epsilon$ uniformly converges to $\Psi$ as $\epsilon_v \rightarrow 0$.*

## D LEARNING VALUE FUNCTIONS: EXPERIMENTAL DETAILS AND ADDITIONAL RESULTS

### D.1 A BRIEF REVIEW OF THE BASELINE METHODS

To evaluate our models, we aimed to compare across a series of different modern motion planning methodologies. In particularly, we consider perspectives across reinforcement like methods such as VIN and IEF, operator learning architectures such as FNO and DAFNO, and physics informed approaches in NTFields and P-NTFields. In this section, we use a series of different GPUs for training and testing. To clarify, all the 2D experiments use a NVIDIA 3090 Ti. The 3D iGibson experiments use a NVIDIA $A100$ for data-generation and training, while we employ a NVIDIA 4060 during testing.

- **Value Iteration Networks (VIN)** Tamar et al. (2016) VIN attempts to learn value functions by approximating the value iteration (VI) algorithm. To do so, they introduce two neural network components to their algorithm. The first model, deemed the VI module, takes in the previous value function estimate and the current reward observed to provide an estimate on the current value function. In implementation, for 2D experiments, the VI module is a classical CNN. They then combine this module a planning module that takes in the value function and current state and passes this through an attention mechanism coupled with a classical NN to achieve the best path direction.

- **Implicit Environment Functions (IEF)** Li et al. (2022) IN IEF2D/IEF3D, the authors take the classical neural implicit network architecture, which represents 2D and 3D scenes via their signed distance functions, and instead learns to represent the scene by obtaining distances between samples start goal pairs. As such their methodology is continuous, and is able to learn trajectories by analyzing these distance functions. However, to generalize across environment's, the authors first project the scene to a latent space via a auto-encoder and then learn a neural implicit function in that latent space.

- **Fourier Neural Operators (FNO)** Li et al. (2021) Like PNO, FNOs learn the solution of operator mappings across continuous function spaces. In particular, FNOs project the input map into a high dimensional latent space, of which an FFT is then performed. From here, the FNO learns a weight matrix in frequency space that multiplies with the encoded input before performing an IFFT and then reprojecting back to the desired output resolution. They have been extremely successful in learning operator solutions specifically in the context of weather (See Kurth et al. (2023)).

- **Domain-Agnostic Fourier Neural Operators (DAFNO)** Liu et al. (2023) DAFNO, an extension of FNO, maintains the same network structure as an FNO, but additionally encodes the domain geometry into the kernel calculation in frequency space. That is,

DAFNO, multiplies the encoded input in frequency space by an additional mask containing the domain geometry. This approach, taking advantage of the fact that Fourier transforms only operate on periodic grids, enable users to learn FNO approximations on non-uniform geometries.

- **Neural Time Fields (NTFields)** Ni & Qureshi (2023a) NTFields is similar to the operator learning architectures in that the network aims to learn an operator, but specifically they constrain that operator to be the solution operator of an Eikonal PDE. In particular, they re-parameterize the value function as a factorized Eikonal equation $V(\boldsymbol{x}, \boldsymbol{g}) = \frac{\|\boldsymbol{x}-\boldsymbol{g}\|}{\tau(\boldsymbol{x}, \boldsymbol{g})}$ where $\tau$ is learned via a neural network. They then perform training with a physics informed loss based on the satisfaction of the Eikonal PDE enabling NTFields to learn paths *without training data*. As such NTFields is able to generalize to high dimensional environments, but is constrained as it needs retraining for each new environment geometry.

- **P-Neural Time Fields (P-NTFields)** Ni & Qureshi (2023b) P-NTFields is an extension to NTFields where the authors propose a progressive learning approach. Particularly, they modify the model training of NTFields by linearly increasing the speed field as the model is trained for more epochs. This helps eliminate the local minima in the value function and thus better captures the geometry of the environment. However, like NTFields, P-NTFields is a physics informed approach and therefore is limited as it also requires retraining for each new environment geometry.

### D.2 SMALL-SCALE 2D MAZE EXPERIMENTS

To compare with state of the art value function predictors such as VIN and IEF2D, we explore the efficacy of a PNO on the small-scale 2D Grid-world experiments in Tamar et al. (2016). We used the same dataset used to test the VIN framework of which IEF2D also employs.

The parameters and the relative errors of our model are presented in Table 4. To develop our model, it took approximately 40 minutes of training on a NVIDIA RTX 3090 Ti GPU.

Table 4: Model parameters and performance metrics for PNO over the GridWorld dataset. The PNO for $8 \times 8$ was smaller than that trained for $16 \times 16$ and $28 \times 28$.

| Map size | PNO model number of parameters | Avg. $L_2$ relative error training data | Avg. $L_2$ relative error test data |
|---|---|---|---|
| $8 \times 8$ | 238816 | 0.019 | 0.022 |
| $16 \times 16$ | 690432 | 0.031 | 0.035 |
| $28 \times 28$ | 690432 | 0.027 | 0.045 |

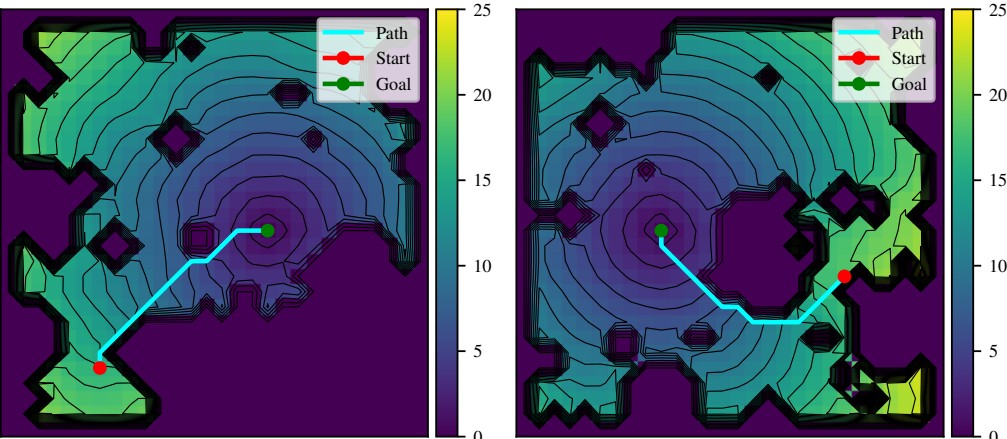

Figure 6: Two examples of PNO planning on $28 \times 28$ Grid-world dataset. Planning is done on the value function generated by PNO using classical gradient descent.

### D.3 MOVINGAI CITY EXPERIMENTS

In this experiment, we highlight the super-resolution capabilities of the planning operator architecture. To do so, we created a custom $64 \times 64$ resolution dataset (will be publicly available) and trained the

neural operator on the custom dataset. Note, this dataset consists of just objects on a rectangular grid as shown in the example in Fig. 2. Furthermore, our dataset did not include any of the city maps, but the synthetic object maps closely resemble a similar structure as the city maps. Our architecture consisted of $157808$ parameters and we achieved an $L_2$ relative error of $0.1$ for both training and testing taking approximately 10 minutes for training on a NVIDIA 3090 Ti GPU.

To showcase our the benefit of both the obstacle encoding and inductive bias, we perform comparisons across a wide set of operator learning benchmarks. We briefly summarize both the training and testing error in Table 1. Note all the testing sets in each category are the same maps structurally, just scaled to the target resolution via averaging. For the in-distribution obstacle dataset, we consider $1k$ maps with 10 different goals each for training. and 10 maps with 10 different goals each for testing. For the MovingAI city experiments, we include all 30 city maps with 3 goals each for testing. An example of the MovingAI city result along with comparison errors is given in Fig. 8 for a snapshot of Paris. Additionally, we show the effect of the PINN loss in Fig. 7 which certainly improves the gradient's residuals. Furthermore, to all models were trained and tested using the PyKonal FMM for the value function and a classic numerical solver for the SDF was used for DAFNO. For PNO, the FNO generated SDF was trained over 1k instances of the $64 \times 64$ dataset (achieving an relative $L_2$ error of $0.068$) and is also performing super-resolution when applied.

Lastly, for the same calculations, we compute the speedup of the neural operator at various resolutions on a NVIDIA 3090 Ti GPU (ML models) and utilize the extremely powerful AMD Ryzen 9 7950X CPU for the numerical solvers. In Table 5, we break down exactly the calculation time of both the SDFs as well as the value function where on small scaled maps, it is clear the numerical solver performs best; however, as the scale increases, we can see that the operator architectures do not lose much computation expenditure while the numerical solvers scale poorly.

Table 5: Computation times for super-resolution calculations average over 1000 instances on the Moving AI lab $2D$ dataset ($64^2$ indicates $64 \times 64$). The DAFNO SDF was calculated using the SciPy numerical solver. The numerical solver used for FMM is via Pykonal.

| | Avg. computation time signed distance function(1000 inst, ms) ↓ | | | | Avg. computation time value function(1000 inst, ms) ↓ | | | | Avg. computation time total function(1000 inst, ms) ↓ | | | |
|---|---|---|---|---|---|---|---|---|---|---|---|---|
| Map size | $64^2$ | $256^2$ | $512^2$ | $1024^2$ | $64^2$ | $256^2$ | $512^2$ | $1024^2$ | $64^2$ | $256^2$ | $512^2$ | $1024^2$ |
| FNO | — | — | — | — | 1.6168 | **1.6356** | **1.6563** | **2.281** | 1.6168 | **1.6356** | **1.6563** | **2.281** |
| DAFNO | **0.1734** | 1.9806 | 8.4207 | 46.1614 | 2.9232 | 2.9889 | 3.024 | 3.6626 | 3.0966 | 4.9695 | 11.4447 | 49.825 |
| PNO | 1.6168 | **1.6356** | **1.6563** | **2.281** | 3.954 | 3.6729 | 4.6735 | 6.0332 | 5.5708 | 5.3085 | 6.3298 | 8.3142 |
| Numerical solver (FMM) | — | — | — | — | **0.3454** | 5.9612 | 25.6555 | 104.6184 | **0.3454** | 5.9612 | 25.6555 | 104.6184 |

## D.4 3D IGIBSON DATASET

For our 3D experiments, we use the iGibson Dataset Shen et al. (2021). The dataset consists of only 10 maps which are not sufficient to train an accurate model. As such, we perform two augmentations for generating sufficient training data. First, we consider different training instances by rotating the different maps across about the z-axis by 90 degrees creating four versions of the same map. Additionally, we augmented the dataset with 32 maps from the HouseExpo dataset Li et al. (2020). The maps were extruded to form 3D maps. Then, for each map, we randomly sampled 5 start goal pairs leading to a total dataset size of $360$. We then performed a $90/10$ train test split explicitly ensuring that the training dataset did not contain any Samuel environments. Further, no start goal positions in the evaluation dataset in Fig. 3 were seen during training. Fig 3 shows the best performing $L_2$ example of our result. For completeness, we also present the worst performing example of our approach in Figure 9. Our model consisted of $418528$ parameters of which we achieved a $0.08$ $L_2$ relative training error and a $0.19$ $L_2$ relative testing error for learning the value function. The computational calculations for value function generation were computed using a NVIDIA 4060 GPU.

## D.5 PLANNING WITH A 4 DOF MANIPULATOR

For our 4-DOF Manipulator experiments, we generate our dataset by randomly positioning the obstacles in the workspace of the robot. We choose the OpenManipulator-X (RM-X52-TNM) to test our methods. For training our PNO, we generate the binary occupancy map in the configuration space of the robot of the size $17^4$. We use 40 maps with 10 randomly generated goal positions each to generate our training data. The test set used 10 maps with 10 randomly generated goals each. To

generate the binary occupancy map, we use the checkCollision function available in MATLAB for each state in the state space of the manipulator. Our model consisted of 55132 parameters. In terms pf performance, we achieved a $0.027$ $L_2$ relative training error and a $0.041$ $L_2$ relative testing error. The model was trained and tested on a NVIDIA 3090Ti GPU. The plan generated by our method was simulated and rendered on MATLAB. 10 shows a slice of of the value functions generated in the configuration space with Joint 3 and Joint 4 fixed at $0°$. The Binary occupancy map generation took 5 hrs for 50 maps with randomly positioned obstacles. For training, FMM was used and the data generation took 40s while the model took 20 min to be trained over 200 epochs.

## E    EMPLOYING OPERATOR LEARNED VALUE FUNCTIONS AS NEURAL HEURISTICS: EXPERIMENT DETAILS AND ADDITIONAL RESULTS.

For this experiment, we trained directly on the city maps at $256^2$, $512^2$ and $1024^2$ resolutions. All of the training and testing in this section was completed using an NVIDIA $A100$. To build our dataset, we considered the 30 city maps provided by the MovingAI city dataset along with 10 goals for each map. Additionally, we randomly erode each of the maps between 1 and 30 layers to generate more data and to help the operator infer the effect of erosion. Again, for each eroded map, we use 10 goals.

For employment as a heuristic, FMM was insufficient for generating training data. This is due the fact A* algorithm works using a set of discrete control inputs to find a path with the shortest distance. However, the value function generated by the Eikonal equation (FMM) represents a value function for a continuous control input space. Using this directly as a heuristic does not yield an accurate value function since such a function largely under approximates the cost-to-go function for the shortest distance problem

Table 6: Model parameters and performance metrics for the PNO models trained and deployed as neural heuristics.

| Map size | PNO model number of parameters | Avg. $L_2$ relative error training data | Avg. $L_2$ relative error test data |
|---|---|---|---|
| $256 \times 256$ | 26029 | 0.051 | 0.065 |
| $512 \times 512$ | 102048 | 0.049 | 0.055 |
| $1024 \times 1024$ | 161920 | 0.058 | 0.053 |

with discrete control space. In order to alleviate this issue we train our neural operator on value functions generated using the Dijkstra Algorithm that gives the value function at every node for the same set of control inputs. This required a data generation time of $8$ hours for the $1024 \times 1024$ city maps (3 hours and 25 minutes for the $512 \times 512$ and $256 \times 256$ maps), but only needed to be complete once, offline. We provde the full model sizes as well as the relative errors in Table 6.

In addition to our results in Section 5, we also conducted an analysis on the effect of erosion. Fig. 11 shows that as the amount of layers eroded increases, the path improves at the cost of expanding more nodes as the heuristic which is as expected given that a fully eroded map yields the Euclidean norm. Perhaps future work can explore different eroding methods for improving the admissibility of the value function.

Value functions and relative error between FMM (Unseen)

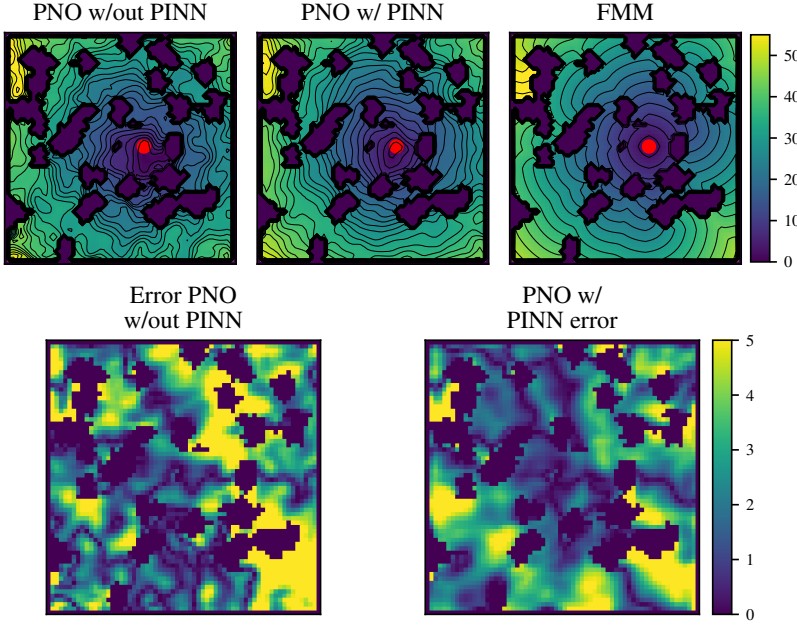

Gradient of value function ($\|\nabla V(x)\|$) and error
of gradient value function ($|\|\nabla V(x)\| - c(x)|$) (Seen)

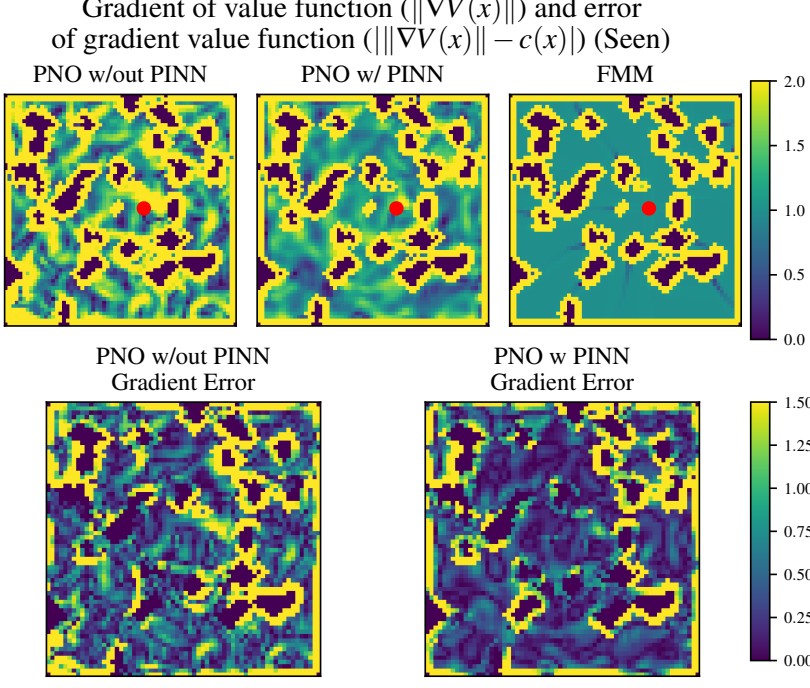

Figure 7: Comparison of PNO with and without the hybrid PINN loss. The left image shows the value function and corresponding error while the right image shows the gradient norm $\|\nabla V\|$ and the corresponding gradient norm error. The example given is in the test set from the $64 \times 64$ synthetic maps and the red dot indicates the goal position. For this example, the model was trained with $\xi = 0.05$ in (15).

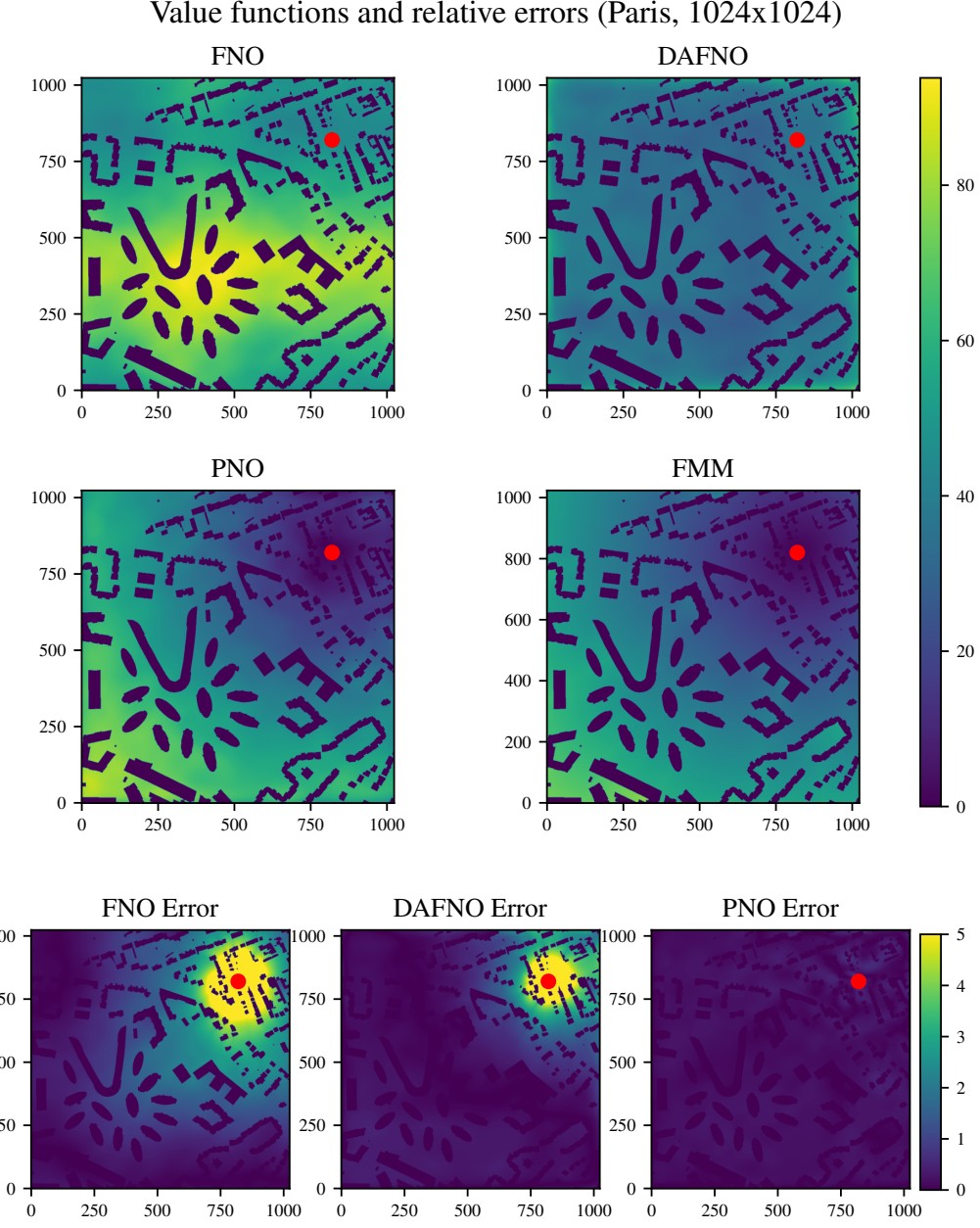

Figure 8: Example of various operator architectures on a Paris $1024 \times 1024$ map. The red dot indicates the target goal position.

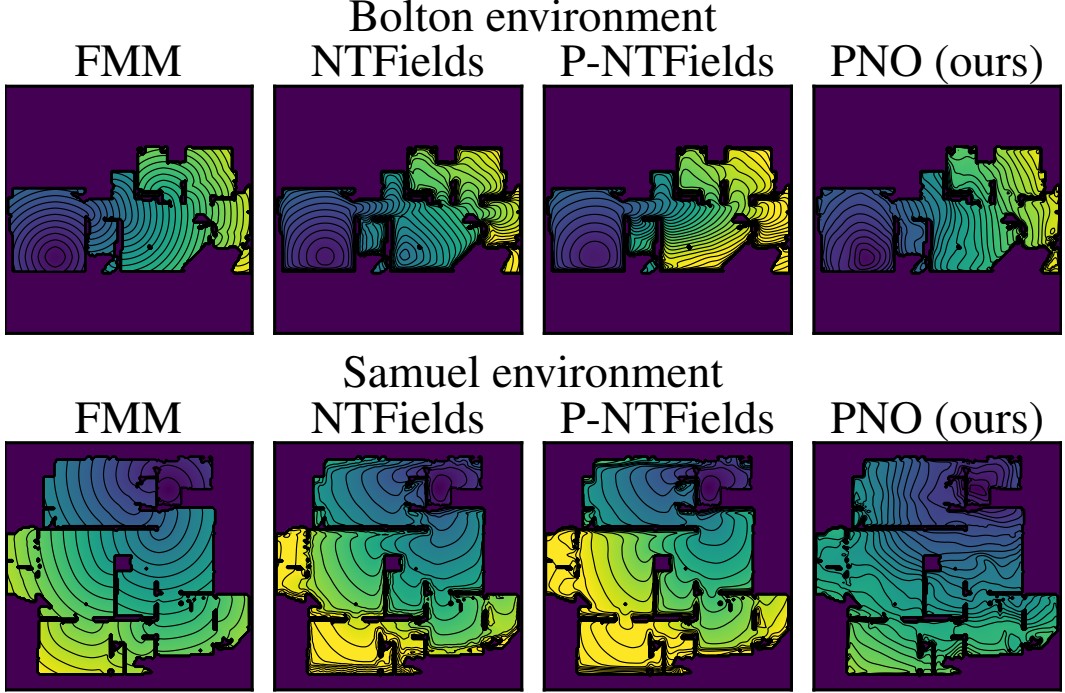

Figure 9: A second example of comparison between NTFields, P-NTFields, FMM and PNO unseen during training. This example is the worst-performing example for the PNO in terms of $L_2$ loss.

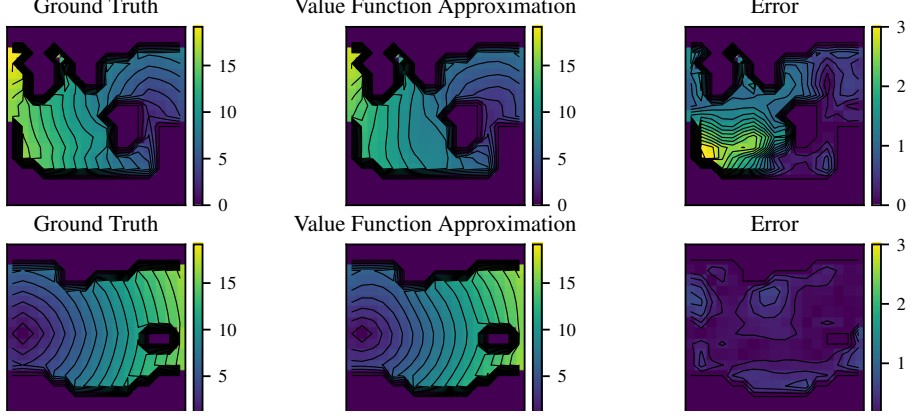

Figure 10: Value function approximation by PNO in 4D C-Space for examples shown in Fig. 4. The value functions are visualized as slices, with Joint 3 and Joint 4 fixed at an angle of $0°$.

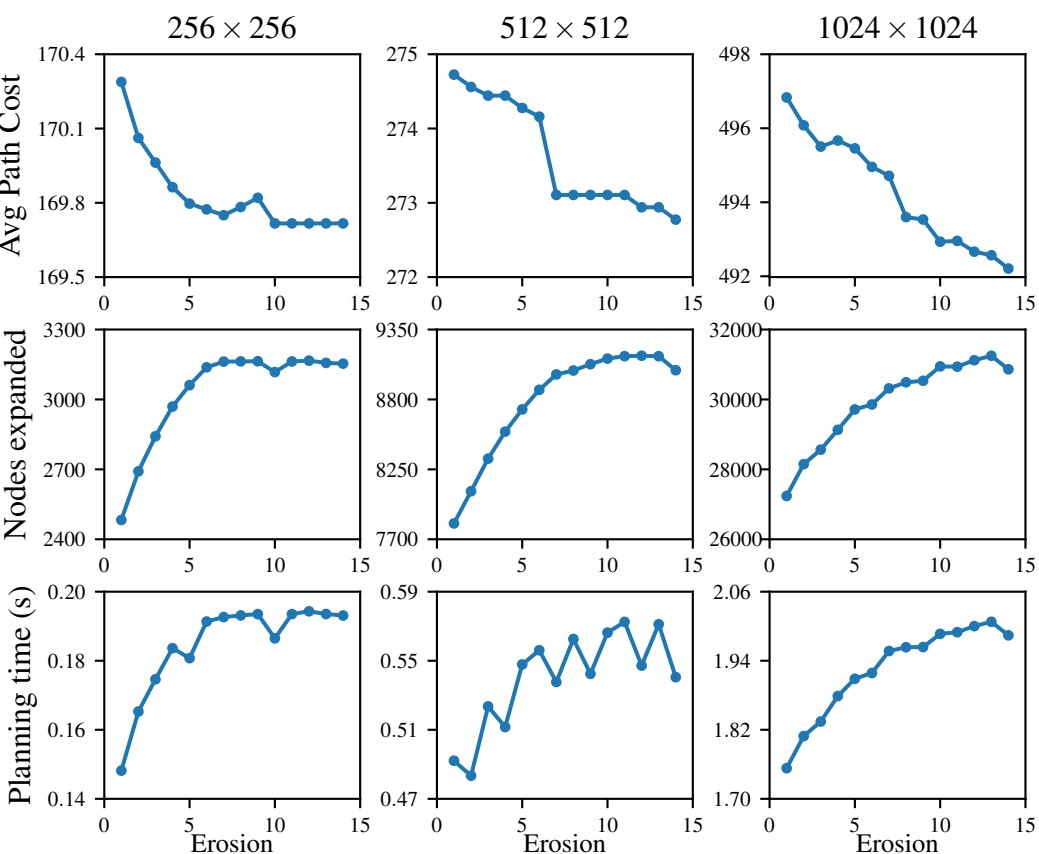

Figure 11: Effect of erosion on various parameters

