# OpenReview forum: "Generalizable Motion Planning via Operator Learning"
_ICLR.cc/2025/Conference — ICLR 2025 Poster_

### Official Review · Reviewer_hZWb · 2024-11-02

**Soundness:** 2
**Presentation:** 3
**Contribution:** 2
**Rating:** 6
**Confidence:** 3

**Summary:**

This paper proposes a method, called Planning Neural Operator (PNO), for approximating the value function of a motion planning problem, viewed as the solution operator to the Eikonal PDE. The authors claim to achieve generalizable motion planning by using a resolution-invariant architecture, which encodes obstacle geometries as well as goal positions. This is done by leveraging the Fourier Neural Operator (FNO) and the Domain-Agnostic FNO (DAFNO). The paper also proposed a method for integrating the learned operator as a heuristic into the A* algorithm which achieves a 33% decrease in the number of explored nodes while maintaining epsilon-consistent optimality.

**Strengths:**

- The paper, although math-heavy, was an enjoyable read. The authors did a very good job of presenting complex theoretical concepts simply while maintaining a good formalism.
- The authors provide a detailed literature review and clearly position the proposed approach within existing work  in traditional and neural motion planning.
- Experiments are conducted on multiple datasets, showcasing the performance of the method on different types of 2D and 3D environments of different resolutions. Results support most of the authors’ claims.

**Weaknesses:**

- Several assumptions are made to justify the existence of the neural operator approximation. However, the reasoning behind these assumptions, and their implications in practical, real-world scenarios, are not thoroughly discussed.
- The model’s performance for robots with high degrees of freedom (DOF) such as manipulators are not explored. This raises questions about the applicability of the approach to more complex, higher-dimensional tasks common in robotic motion planning.
- While the paper mentions design decisions like encoding obstacle geometries and ensuring triangle inequality, the specific impact of these features is not thoroughly examined through ablation studies. It would be beneficial to quantitatively show how each architectural component contributes to the final performance, especially the modifications to the Fourier Neural Operator (FNO). Also, comparing the use of DeepNorm to other possible parametrizations could help justify the design choices even more.

**Questions:**

- Could you provide more details on the intuition behind the assumptions required in the paper and how it relates to the practical application of your method in real-world scenarios?
- Have you tested PNO in real-world robotic setups beyond the synthetic and iGibson environments? If so, could you share results or insights on how PNO performs in tasks involving more complex real-world constraints or unstructured environments?
- How does the proposed method scale to tasks with significantly higher degrees of freedom (DOF) beyond the environments presented in the paper? Have you performed any scalability studies on more complex robotic systems?
- Could you provide more detailed ablation studies that show the contributions of specific components in your architecture, such as obstacle encoding, triangle inequality enforcement, and modified Fourier Neural Operator layers? This would clarify how each design choice impacts overall model performance.
- In table 1, do you have an explanation on why the success rate of your method particularly drops for the 16x16 maps ?

---

> ### Author Response · Authors · 2024-11-26
>
> Thank you very much for your constructive and detailed review. Your suggestions have been extremely helpful for improving the quality of the paper. We also appreciate that you found the paper to be an enjoyable read, despite its mathematical details.
> 1. **Clarification of practical applicability of the assumptions:** Thank you for the question on clarification of the assumptions.
>    - For Assumption 1, we require the existence of a proper policy. In practice, this assumption is satisfied when there is a feasible path from every start position to the goal. This assumption excludes trivial planning problems that may have no solution.
>    - For Assumption 2, we require equicontinuity and uniform positivity for the space of possible cost functions. In practice, this assumption can be satisfied by creating smooth boundaries around each obstacle and associating them with strictly positive motion cost.
>
>    We added sentences after both of these assumptions to clarify the practical meaning of the assumptions.
>
> 2. **Scalability and more challenging environments:** We thank the reviewer for their suggestion to utilize PNO in more complex environments.
>    - We emphasize that by planning in continuous function space, our method is able to scale to large $2D$ and $3D$ environments. For example, there has been very little work that considers neural motion planning on environments of the size $1024 \times 1024$ which consists of $1$ million+ points. However, as the reviewer points out, this task is not necessarily the most complex because it assumes a point robot. Thus, to highlight the ability of our approach to handle scenarios with more complex robot configurations, we introduced a $4$-DOF manipulation task within the short rebuttal period. We show that our PNO architecture is able to generate viable value functions for manipulator motion planning in Figure 8 and highlight to examples of end-effector trajectories in Figure 4. We believe that extending the PNO to $4$-DOF manipulators in a short time shows the strong promise of the PNO design to handle high DOF robot systems. We acknowledge that additional improvements are needed in our architecture to eliminate the discretizaiton of the input cost function and enable scaling to high DOF configuration spaces. We think that the $4$-DOF experiments are encouraging for extensions to $6$+ DOF manipulators.
>
>     - We leave implementation on real-world robots as future work, given that in this paper we focus on the learning architecture design. Particularly, this work introduces  new connections between the *neural operator* approach and motion planning problems. Thus, we present an analysis both theoretically and pragmatically exploring the generalization and super-resolution capabilities of the general learning algorithm across a variety of different tasks (2D planning in real city maps, 3D iGibson navigation, 4D robot manipulator planning, and deployment as a neural heuristic in classical motion planning algorithms).
>
> 3. **Ablation study:** We intended for the MovingAI city maps to highlight the architecture advantages of the triangle inequality preservation and obstacle encoding in our model. To clarify this, we have made multiple additions to the Moving AI city maps section and highlighted the effects of the PNO in Table 2. In Table 2, we see that the improved performance of obstacle encoding when comparing FNO (without obstacle encoding) to PNO. Further, we highlight the generalization improvement of the triangle inequality output layer in the same Table when comparing PNO and DAFNO (the latter does not include the triangle inequality). We thank the reviewer for helping us better showcase the advantages of our approach with this suggestion.
>
> 4. **Table 1:** We identified the issue with the results on the Grid-World dataset. The poor performance of our method is attributable to the FMM solver, which computes value functions in continuous space. States that are laterally surrounded by obstacles in the x and y directions become effectively unreachable because they can only be accessed diagonally while grazing the obstacles—a scenario that the FMM solver does not permit as it is considered a collision. Thus, the PNO model, which is trained using the FMM solver outputs as labels is unable to correctly generate paths when the start and goal positions require a diagonal movement between two touching obstacles. These types of maps are most prevalent in the $16\times 16$ dataset and actually are one of the major issues stopping PNO from reaching a higher success rate in all three map sizes. If one trained our PNO model with Dijkstra generated labels that account for these diagonal movements, the issue would be alleviated. However, to stay consistent, we did not modify the FMM solver nor the maps to ensure a fair comparison with the other benchmarks.

---

### Official Review · Reviewer_6Xqc · 2024-11-03

**Soundness:** 3
**Presentation:** 2
**Contribution:** 3
**Rating:** 6
**Confidence:** 4

**Summary:**

This paper introduces the novel use of neural operators to solve the Eikonal equation for motion planning in 2D and 3D environments. The proposed method takes an environment occupancy map, along with start and goal points, to predict the optimal value function for path planning in free space. By leveraging the generalization capabilities of neural operators, the authors demonstrate improved adaptability across different environments compared to conventional methods. Experimental comparisons with search-based methods and neural motion planners validate the effectiveness and computational efficiency of this approach.

**Strengths:**

1. Innovative Use of Neural Operators: The authors adopt neural operators, which have shown superior generalization in PDE-solving applications, suggesting potential advancements in both performance and scalability.

2. Generalization Across Datasets: Experimental results indicate the method’s potential to generalize across diverse environments, which is a noted advantage over physics-informed neural networks (PINN) approaches.

3. Computational Efficiency: The approach demonstrates fast computational performance for small-resolution cases, making it potentially suitable for real-time applications.

**Weaknesses:**

1. Scalability Constraints:
Since the method relies on grid-space convolutions, it faces intrinsic limitations in scaling to higher-dimensional spaces, a notable restriction for applications in manipulation tasks.

2. Dependence on Supervised Learning:
 The method’s reliance on supervised learning demands ground truth PDE solutions, which may be impractical to obtain in complex environments.

3. Presentation Issues:
 - Notation Clarity: The pipeline figure uses the same symbol for the SDF operator and PNO, leading to ambiguity. It would be clearer to differentiate these in the figure.
- The neural network structure lacks clarity. The description mentions $F^{-1}(R(F()))$, but it may be more informative to explicitly state the intermediate layers, e.g., $L = F^{-1}(R(F()))$, with a clear distinction that the output should be denoted as $\Phi $ rather than $L_M $.

4. Experimental Comparisons:
 The paper currently compares against VIN and NTFields, which are established but not necessarily state-of-the-art. Incorporating comparisons with more advanced methods like Differentiable Spatial Planning using Transformers [1] and Progressive Learning for Physics-Informed Neural Motion Planning [2] would provide a more comprehensive benchmark.

[1] Differentiable Spatial Planning using Transformers

[2] Progressive Learning for Physics-Informed Neural Motion Planning

The original experiment struggled to fit the value function accurately, but adding the PINN loss has shown improvements. However, the current method has only been tested on environments with isolated cluttered obstacles. This makes it difficult to assess its ability to generalize to more complex scenarios, such as the maze-like Gibson environments, where optimal paths often require navigating through multiple turns.

**Questions:**

Despite its limitations in scaling and reliance on ground truth training data, the proposed method represents a promising direction for motion planning through neural operators. I have the following questions to further explore its potential:

1. Pure PINN Loss: Could pure PINN loss be used to train the network instead of supervised learning? An analysis of its generalization performance could reveal insights into the potential for unsupervised or semi-supervised learning.

2. Testing on General Geometries: The method’s scalability could be further assessed by testing on general geometries, as seen in [1]. This could help to understand whether the approach can extend to higher-dimensional space planning.

3. Visualization of Results: It would be more effective to display contour lines of the value function instead of only color representations. Contour lines could better reveal any artifacts in the value function, providing clearer insights into the method's performance.

[1] Fourier Neural Operator with Learned Deformations for PDEs on General Geometries

---

> ### Author Response · Authors · 2024-11-26
>
> Thank you for your constructive and detailed review. Your suggestions have been extremely helpful for improving the quality of the paper.
>
> 1. **Pure PINN loss:**  A key advantage of our theoretical formulation is that it encompasses a wide class of operator networks including physics informed neural operators (PINO, [1]). As such, one can certainly employ a pure PINN loss to train a network without data labels under our framework. This would enable discretization-invariant training without supervised labels but would most likely sacrifice the generalization as PINNs typically *require retraining* for every new environment. Moreover, it is also possible to use a hybrid approach. Namely, instead of a pure PINN loss, one can utilize a combination of the supervised approach with a PINN loss correction term, which has been explored for PDEs in [1]. This would of course require labeled data, but perhaps only for a few environments which is then achievable to compute - In fact, one could also perhaps use neural network generated labels from P-NTFields/NT-Fields to create these labels paying a large upfront cost offline in exchange for computationally fast deployment in a variety of environments online.
>
> 2. **Testing on general geometries:** We very much appreciate this suggestion as it highlights the power of connecting robot motion planning with operator learning in that we can take advantage of the existing literature that has been successful in the SciML community. However, one challenge arises in that our methodology encodes specific properties (i.e., obstacles and triangle inequality) in the value function. It is not obvious how those properties would be enforced if the input requires a latent space representation via an autoencoder along the lines of references [1] mentioned by the reviewer. This is an exciting future direction to consider.
>
>    We can, however, work directly on general geometries by circumscribing the geometry with a rectangle and then using $\tilde{\chi}_{PNO}$ as in the PNO framework to treat the points not in the geometry as obstacles. With this approach, one can then apply PNO on any desired geometry.
>
> 3. **Presentation:** We made the following revisions to improve the presentation.
> - We added contour lines to the value functions in Figure 3.
> - Additionally, we added sentences throughout Section 3 to describe the components of our architecture mathematically. We thank the reviewer for helping us improve the clarity and therefore the usefulness of our approach.
>
> 4. **Baselines:** As addressed above, we added comparisons with P-NTFields (See above and Figure 3 of the revised manuscript).
>
> [1]  Z. Li, H. Zheng, N. Kovachki, D. Jin, H. Chen, B. Liu, K. Azizzadenesheli, and A. Anandkumar. Physics-informed neural operator for learning partial differential equations. ACM/IMS J. Data Sci., 1(3), May 2024

---

> > ### Comment · Reviewer_6Xqc · 2024-11-27
> > **Thank you for your response**
> >
> > Thank you for your thoughtful feedback and for sharing updated results. I appreciate the effort in presenting these findings but would like to address a few concerns.
> >
> > 1. **Artifacts in Learned Value Function:** In the Gibson environments, the learned value function’s contour lines show noticeable artifacts compared to P-NTFields based on FMM. While L2 error is useful, it does not fully reflect motion planning quality. I suggest evaluating success rates in motion planning tasks for a more practical assessment.
> >
> > 2. **4D Grid-Based Manipulator Experiment:** The reliance on discrete grids leads to the curse of dimensionality, making this experiment less compelling unless alternatives like continuous representations are explored.
> >
> > 3. **Fit to Eikonal Equation:** The method seems to struggle with accurately fitting the discontinuous Eikonal equation solution. I recommend exploring continuous Eikonal equation formulations or using physics-informed losses to address this limitation.
> >
> > While the proposed direction is promising, the current results are not fully convincing. I encourage further refinement and broader experiments to strengthen the work’s impact.

---

> > > ### Author Response · Authors · 2024-11-29
> > >
> > > Thank you for your response and additional questions! We appreciate your interest in the work as well as your insightful suggestions. To clarify these aspects, we explain our methodology as well as design decisions in more detail below.
> > > 1. **Artifacts in Value Function:** We agree that, compared to FMM or P-NTFields, there are artifacts in the value function approximation. However, we want to emphasize that our method is not just working on a single environment but across many different environments (and goal configurations) without retraining. Thus, given our approach is able to approximate the value function for *any* input geometry compared to just one,  it is fair to expect some artifacts in the contour lines compared to P-NTFields.
> > > 2. **Reason for** $L_2$ **error:** We want to emphasize the reasoning for our choice to use the average relative $L_2$ error over success rate. The main reasoning is that success rate is heavily influenced  by the gradient descent algorithm chosen (e.g., classical GD, GD with momentum, bi-directional GD, use as heuristic in motion planning, etc.) and various papers will use various forms of GD that can highlight the advantage of their approach. In contrast, the $L_2$ error of the value function is a universal metric that can be compared fairly across a variety of approaches without modification.
> > > 3. **Success Rate and Section 5:** Given the reviewer’s question about success rates, we want to emphasize the importance of Section V for addressing this. In particular, we believe that, in planning for real-world problems, anything less than 100% success is not acceptable due to safety of both the robot and other entities in the environment. Thus,  we pursued the development in Section V which shows that the value function predictions can be used as heuristics to accelerate motion planning  while *guaranteeing* 100% success rate compared to an approach like GD on the learned value function directly. Further, we show theoretically that this architecture guarantees an $\epsilon$-consistent heuristic giving the user a guarantee that the path computed not only is successful, but is also $\epsilon$-optimal.  With this approach, we still obtain the ultra-fast advantage of a neural-operator method while guaranteeing that there are no local minima that can destroy the path success rate.
> > > 4. **Learning in Continuous vs Discrete Space:** With respect to the manipulator, we want to clarify that the *learning* is in continuous space - namely Fourier space. Thus, one can project our result to any resolution of interest, as we highlight in the MovingAI city maps.  Our approach only suffers from a curse of dimensionality due to the input representation but we emphasize that one can use an autoencoder with our approach to avoid this curse since the approach itself learns in continuous space.
> > > 5. **Fit to Eikonal Equation:** We thank the reviewer for pointing out this extension and we certainly agree that the fit to the Eikonal equation can be improved using a physics informed loss. We are looking forward to exploring this direction in future work.

---

> > > > ### Comment · Reviewer_6Xqc · 2024-11-29
> > > > **Thank you for your response**
> > > >
> > > > Thank you for your detailed response and explanations. While I appreciate the effort to address my concerns, I still find the current explanations insufficient for the following reasons:
> > > >
> > > > 1. **Generalization and Value Function Quality:** The learned value function resembles Euclidean distance rather than the solution to the Eikonal equation. This undermines the claim of achieving generalization. Additionally, P-NTFields do not rely on the ground truth value function for training, supervised learning methods should be expected to perform similarly.
> > > >
> > > > 2. **Comparison to Euclidean Distance:** From the contour plots, the learned value function seems closer to Euclidean distance than the true Eikonal solution. I would like to know if the L2 error to Euclidean distance is smaller, as this would clarify the degree of deviation from the expected solution.
> > > >
> > > > 3. **Heuristics for Planning:** Any distance function can serve as a heuristic for A* or other search methods to guarantee an optimal solution. However, the goal of solving the Eikonal equation is to provide better heuristics, and the current results fall short in demonstrating this advantage.
> > > >
> > > > 4. **Computational Complexity:** While the learning occurs in Fourier space, FFT is still required to convert between spaces, and its computational complexity grows with the grid's dimensionality. This aspect should not be overlooked when discussing scalability.
> > > >
> > > > In conclusion, while the proposed approach explores a promising direction, the current results and explanations do not sufficiently support its claims. I believe stronger results are needed to justify publication.

---

> > > > > ### Author Response · Authors · 2024-11-30
> > > > > **Thank you for your response**
> > > > >
> > > > > Thank you for your response. Please allow us to address your new concerns regarding
> > > > > the closeness to Euclidean distance.
> > > > >
> > > > > We have developed the following table showcasing the difference in the average L2 relative
> > > > > error for each test environment over 100 start goal configurations.
> > > > > |           |                      Bolton Environment                     |                      Samuel Environment                     |
> > > > > |:---------:|:-----------------------------------------------------------:|:-----------------------------------------------------------:|
> > > > > |           | Avg. $L_2$ relative error b/t Euclidean and  given approach | Avg. $L_2$ relative error b/t Euclidean and  given approach |
> > > > > | PNTFields |                            0.5894                           |                            0.7920                           |
> > > > > | NTFields  |                            0.4655                           |                            0.7095                           |
> > > > > | PNO       |                            0.1084                           |                            0.1892                           |
> > > > > | FMM       |                            0.1561                           |                            0.3309                           |
> > > > >
> > > > > As the reviewer points out, the PNO is closer to Euclidean distance than both PNTFields
> > > > > and NTFields. However, note that, FMM (which we consider to be the ground truth correct
> > > > > value function) is also much closer to Euclidean distance and thus it is expected that our
> > > > > approach is closer to the Euclidean distance compared to NTFields since it is trained with
> > > > > the FMM based labels.
> > > > >
> > > > > With regards to the reviewer Point 3 about neural heuristics, in Section V, we highlight the clear advantages of the
> > > > > PNO generated heuristic over the Euclidean norm. We see that on large grids (1024 × 1024)
> > > > > the PNO generated heuristic reduces nodes expanded by 30% and similarly reduces the
> > > > > computational time as well. This study shows (1) that the PNO is *not learning* the Euclidean
> > > > > norm as the reviewer suggests otherwise we would not see the decrease in nodes expanded
> > > > > and (2) that the advantage of the PNO generated heuristics is significant as 30% is a large
> > > > > reduction in computation when scaled.
> > > > >
> > > > > Finally, in regards to the latest response from the reviewer, we ask the reviewer to consider a full evaluation of our work and believe it is not objective to minimize its contribution to just the accuracy of value function approximation. While this is indeed the ultimate goal of value function prediction, our work establishes two fundamental ideas that may impact subsequent research on generalizable value function learning.
> > > > >
> > > > > (1) We established the first connection between neural operator learning and Eikonal PDE solving in the context of motion planning. We derived the properties required for the existence of accurate neural operator approximators and showed that constraints in motion planning can be encoded explicitly in neural operator learning. Requesting stronger results (such as PINN loss) beyond the 4 experiments given to justify the publication of our work is similar to not giving credit to our paper for this connection and allowing a later publication to claim it.
> > > > >
> > > > > (2) We introduced the key idea to enforce the triangle inequality in the learned value functions. Please note that this is critical to guarantee admissibility and consistency in the predicted value functions, which are required properties for effective use as heuristics in motion planning. Using general approximation methods that do not guarantee admissibility require inefficient modifications of motion planning algorithms, such as revisiting previously expanded states, which lead to significant decrease in efficiency. Claiming that any distance function can serve as a heuristic for A* minimizes the importance of guaranteeing admissibility and consistency in heuristic functions which have been the subject of many impactful studies in heuristic search.

---

> > > > > > ### Comment · Reviewer_6Xqc · 2024-11-30
> > > > > > **Thanks for your response**
> > > > > >
> > > > > > Thank you for the additional experiments and clarifications. I appreciate the effort to address my concerns, but I would like to clarify my point about the PNO results being closer to Euclidean distance than to FMM.
> > > > > >
> > > > > > As shown in the paper's table, in the first scene, the PNO error to FMM is 0.0877, while the error to Euclidean distance is 0.1084. In the second scene, the PNO error to FMM is 0.2362, but to Euclidean distance, it is smaller at 0.1892. This indicates that while the PNO is improving, it does not match the FMM results closely enough, and instead trends closer to Euclidean distance in certain cases. Additionally, the higher error from NTFields and P-NTFields can be explained by the fact that they solve a different Eikonal equation ($||\nabla T||=1/S$ where $S$ is a speed function) compared to FMM and PNO, which solve $||\nabla T||=1$.
> > > > > >
> > > > > > While I agree that the paper highlights a promising direction, the current results indicate that the neural operator is not approximating the Eikonal equation's solution with sufficient accuracy. If the core method cannot achieve highly accurate results, it calls into question the effectiveness of the two key contributions: (1) the connection between neural operators and motion planning, and (2) the triangle inequality enforcement for admissibility and consistency. Without demonstrating strong accuracy in the core value function approximation, the broader implications of these contributions remain unclear.
> > > > > >
> > > > > > I encourage further improvements to strengthen the experimental results, as they are critical to substantiating the claims and contributions of this work.

---

> ### Author Response · Authors · 2024-12-01
> **Thanks for your response**
>
> Thank you for your response and furthermore thank you for your entire discussion throughout this rebuttal period.
>
>
> We have conducted additional experiments as requested by the reviewer showcasing the training of a PNO with a hybrid PINN + relative $L_2$ error loss on the 2D dataset given the short response window. The loss function is explicitly given by Avg $L_2$ relative error +  $\lambda \left(\sum_{x \in \mathcal{S}} (\lVert \nabla u(\mathbf{x})\rVert - c(\mathbf{x}))^2\right)^{\frac{1}{2}}$ with $\lambda=0.05$ as a scaling hyperparameter. Although the PDF is not currently, editable, we promise to include such experiments in the final revision of the manuscript.
>
> We first present the following table for the 2D dataset of which we ensure is extensively detailed for the reviewer as both an ablation study and verification of our approach.
> |                                              |           Avg. relative L2 error $\downarrow$           |         |         |          |                                                                 |         |         |          |
> |----------------------------------------------|:-------------------------------------------------------:|---------|---------|----------|-----------------------------------------------------------------|---------|---------|----------|
> |                                              | $64^2$                                                  | $256^2$ | $512^2$ | $1024^2$ | $64^2$                                                          | $256^2$ | $512^2$ | $1024^2$ |
> |                                              | Synthetic obstacle dataset  (100 maps, in-distribution) |         |         |          | MovingAI real-world city dataset (90 maps, out-of-distribution) |         |         |          |
> | FNO (PNO w/o Deepnorm and obstacle encoding) | 0.1996                                                  | 0.5771  | 0.6214  | 0.6405   | —                                                               | 0.7188  | 0.7519  | 0.7692   |
> | DAFNO (PNO w/o Deepnorm layer)               | 0.0985                                                  | 0.3868  | 0.4060  | 0.4120   | —                                                               | 0.4090  | 0.4259  | 0.4315   |
> | PNO (ours)                                   | 0.1136                                                  | 0.1197  | 0.1190  | 0.1194   | —                                                               | 0.1748  | 0.1885  | 0.2034   |
> | PNO w/ hybrid PINN loss (ours)               | **0.0698**                                                  | **0.0865**  | **0.0869**  | **0.0872**   | —                                                               |**0.1675**  | **0.1761**  | **0.1842**   |
> |                                              | **Avg. relative L2 error $\downarrow$ + PINN error term** |         |         |          |                                                                 |         |         |          |
> |                                              | $64^2$                                                  | $256^2$ | $512^2$ | $1024^2$ | $64^2$                                                          | $256^2$ | $512^2$ | $1024^2$ |
> |                                              | Synthetic obstacle dataset  (100 maps, in-distribution) |         |         |          | MovingAI real-world city dataset (90 maps, out-of-distribution) |         |         |          |
> | FNO (PNO w/o Deepnorm and obstacle encoding) | 0.2101                                                  | 0.6049  | 0.6510  | 0.6710   | —                                                               | 0.7549  | 0.7892  | 0.8071   |
> | DAFNO (PNO w/o Deepnorm layer)               | 0.1048                                                  | 0.4048  | 0.4246  | 0.4309   | —                                                               | 0.4311  | 0.4489  | 0.4547   |
> | PNO (ours)                                   | 0.1217                                                  | 0.1289  | 0.1282  | 0.1286   | —                                                               | 0.1872  | 0.2006  | 0.2157   |
> | PNO w/ hybrid PINN loss (ours)               |**0.0749**                                                 | **0.0933** | **0.0937**  |**0.0941**    | —                                                               | **0.1797**  |**0.1879**  | **0.2157**    |
>
> Given the results, it is clear the PINN loss does improve the performance of PNO and we very much thank the reviewer for their suggestion. We also see it significantly improves the generalization performance when applied to super-resolution of the PNO.

---

> > ### Author Response · Authors · 2024-12-01
> > **Thank you for your response**
> >
> > Second, we showcase that, as the reviewer has suggested, the PINN loss smoothens the contours in the value function in the following images (https://postimg.cc/xX0dmwM3; shared anonymously). The image shown is an unseen testing map of which the PNO with the PINN loss has an improved $L_2$ relative error in the value function as well as the resulting gradient error is much improved in the bottom figure. We believe this highlights to the reviewer both the power of our framework as we have been able to extend it with a PINN loss in under $24$ hours as well as the ability  of our methodology to accurately approximate value functions while maintaining generalization which we emphasize is *only one part* of our overall contribution. We have no doubt the training under PINN loss can be applied in the $3D$ and $4D$ experiments with the same approach and expect that with more time and better tuning, even better approximations can be achieved.
> >
> > In addition to the experiment above, would like to make the reviewer (as well as the AC) aware of the importance of Section V by pointing out another ICLR 2025 submission "Physics-informed Temporal Difference Metric Learning for Robot Motion Planning" (https://openreview.net/forum?id=TOiageVNru) attempting to solve a similar problem in which a reviewer (7oHd) in that submission asks for experiments with value functions as a heuristic in $A^\ast$. As such, given the interest from other members of the community, we hope the reviewer recognizes the contribution of Section 5, which **guarantees the use of the PNO predicted value functions as $\epsilon$-consistent heuristics** for deployment in popular motion planning algorithms such as A*, RRT, and RRT* (as the results shown in Table 3 and Figure 5)."
> >
> >
> > Finally, as we are coming close to the response deadline, we would like to summarize the discussion for the reviewer. Since the original reviews, we have provided 3 additional experiments including the baseline PNT-Fields, a 4DOF-manipulator experiment as well as the above extension with a PINN hybrid loss all of which highlight the flexibility of the neural operator approach. Furthermore, in the final evaluation of our work, we ask the reviewer to view the work holistically recognizing the many contributions and potential for impact our work has. We introduced a new connection between neural operators and motion planning which led to the development of theoretical guarantees of both universal approximation as well as the consistency and admissibility of our neural heuristic. Further, we then highlighted the applicability of our approach across many experiments achieving both computational speedups over FMM as well as in $A^\ast$. Lastly, we showcased the power of our framework for future impact by quickly developing a PINN experiment which falls completely under the theoretical guarantees aforementioned. As such, we believe the work's introduction of operator learning for motion planning has the potential for significant long-term impact in generalizable motion planning and hope that the reviewer similarly recognizes these contributions.

---

> > > ### Comment · Reviewer_6Xqc · 2024-12-01
> > > **Thanks for your additional experiments**
> > >
> > > Thank you for the additional experiments and clarifications. The new results, particularly the PINN extension, are much more convincing and demonstrate the potential of your framework. I think that incorporating a smooth Eikonal equation ($||\nabla T||=1/S$) could further enhance the experimental results. If possible, I suggest including the best combination of methods (PINN + PNO + smooth Eikonal equation) in the final version of the paper to showcase the full strength of your approach.
> > >
> > > However, I still have reservations about the 4D manipulator experiments. The reliance on a discrete grid representation for FNO limits scalability in higher dimensions. Exploring alternatives, such as DeepONet, could be a more effective solution for handling such cases.
> > >
> > > Overall, I acknowledge the progress demonstrated in the additional experiments and believe these refinements strengthen the paper’s contributions.

---

> > > > ### Author Response · Authors · 2024-12-01
> > > > **Thank you for your response.**
> > > >
> > > > We thank the reviewer very much for their time and efforts in the review as well as their suggestions that no doubt improved the paper throughout the rebuttal phase. We will experiment with the reviewer's suggestion for incorporating a smooth Eikonal equation in the final revision and if there are any additional questions, please let us know.
> > > > Best regards,
> > > > The authors

---

> > > > > ### Comment · Reviewer_6Xqc · 2024-12-02
> > > > > **Further experiments**
> > > > >
> > > > > Thank you again for the additional experiments, and I’m glad to see the PINN loss working effectively. However, I believe that one plot alone is insufficient to fully justify the method. Could you provide results for training on the Gibson environment, including both seen and unseen cases? This would help evaluate whether the approach generalizes well to unseen environments and strengthens the overall argument for its effectiveness.

---

> > > > > > ### Author Response · Authors · 2024-12-03
> > > > > > **Thank you for your response.**
> > > > > >
> > > > > > Dear reviewer,
> > > > > > Thank you for the follow up. We agree that one example plot is not be sufficient to show generalization. We have additional plots from the 2D experiments at https://postimg.cc/K4H9Lsdq. In the image, we show two seen maps (left) and two unseen maps (right). Additionally, we ensure that on each side, one map has a far goal in the corner showing long travel distance (left) and the other map has a centralized goal (right). We hope this shows that the PINN loss is indeed helping with generalization in both goals and environments. Additionally, the aggregate statistics for the reviewer in the Table above also corroborate the generalization shown in the plots above.
> > > > > >
> > > > > > We plan to work on the 3D experiments for the final version of the paper. We want to do them carefully which will take some time. There is not sufficient time to train the 3D model as it takes (1) 4+ hours to train on an A100 (of which we don't currently have available from our university's cluster) and (2) requires tuning to ensure the PINN loss is working effectively. However, we agree with the reviewer that adding this experiment is very important and will do it for the final revision. Given the success of the PINN loss in 2D as well as the demonstrated translation of our approach without a PINN loss from 2D to 3D experiments, we are confident that the PINN loss should translate accordingly.
> > > > > >
> > > > > > We thank the reviewer again for their suggestions and the time and effort they dedicated to help us improve our paper during the rebuttal period! We hope the additional examples strengthen the reviewers positive view of the PINN extension and also the overall potential for impact of our framework!

---

> > > > > > > ### Comment · Reviewer_6Xqc · 2024-12-03
> > > > > > > **Thanks for further experiments**
> > > > > > >
> > > > > > > Thank you for providing the additional plots and for your thoughtful follow-up. I appreciate the effort to demonstrate the effectiveness of the PINN loss in seen and unseen cases. However, I believe the Gibson environment presents unique challenges compared to the current 2D experiments. Its maze-like structure often requires optimal paths with many turns, while the current cluttered environments involve fewer directional changes.
> > > > > > >
> > > > > > > This difference raises a potential concern: while the method may generalize well for environments with isolated obstacles, it might struggle in more complex, maze-like scenarios. I acknowledge the value of the approach, but I believe fully addressing generalization across diverse environments, particularly maze-like ones, remains an open question.

---

### Official Review · Reviewer_C6NW · 2024-11-07

**Soundness:** 3
**Presentation:** 3
**Contribution:** 2
**Rating:** 5
**Confidence:** 3

**Summary:**

This paper casts motion planning as a problem of learning a value function modelled by an Eikonal PDE. The authors propose to view the solution as a non-linear operator mapping costs to value function solution, and learn this using fourier neural operators. The paper proposes to adjust standard Fourier neural operators by (1) encoding the obstacles, given by an SDF, and (2) enforce triangle inequality is satisfied. The paper argues that the neural operator formulation enables the querying to be done in at a higher resolution than the resolution of 2D maps used in training.

I appreciate the novelty of casting motion planning into an operator learning problem, and the approach presented is solid. My main concern is a critical one: the simplicity of the experimental setups -- the evaluations were all conducted in 2d "grid-world"-like environments (point mass moving in 2d layouts e.g. Fig 4.), whereas other methods of learning how to plan generally evaluate on higher degrees of freedom tasks, such as robot manipulators of 6 or 7 DOF.

**Strengths:**

See review above

**Weaknesses:**

See review above

**Questions:**

How well does the learned planner scale wrt the dimensions of the search space? As most of the citations come from the robotics literature, I would like to see motion planning beyond 2D or 3D toy examples. Consider evaluation on robot manipulators with 6 or 7 DOF.

---

> ### Author Response · Authors · 2024-11-26
>
> Thank you very much for your constructive and detailed review. We appreciate your suggestions for improving the paper! As aforementioned, we added a planning experiment on a $4$-DOF robot manipulation task in Figure 4 and Figure 8. We believe that extending the PNO to 4-DOF manipulators in the short rebuttal period shows the strong promise of the PNO design to apply to high DOF motion planning problems. In the future, we plan to extend the PNO architecture to real-world $6$ and $8$ DOF manipulators. We acknowledge that the current cost function representation needs to be generalized to achieve scalability in higher DOF problems. One approach would be to invoke an autoencoded representation of the geometry of which neural operator approaches exist as in [1]. Hence, it may be possible to preserve the discretization invariance property of PNO in this setting but it is not obvious how to ensure the satisfaction of the triangle inequality and the hard obstacle constraints in the latent space. We leave this as an exciting direction for future work.
>
> [1] J. H. Seidman, G. Kissas, G. J. Pappas, and P. Perdikaris. Variational autoencoding neural operators. In International Conference on Machine Learning (ICML), 2023.

---

> > ### Author Response · Authors · 2024-12-01
> > **Thank you for your review.**
> >
> > Dear Reviewer, Thank you once again for your thoughtful review and for raising critical points that have helped us refine our work. We wanted to kindly follow up on our earlier response to your comments, as we are coming close to the response deadline and would greatly value your input.
> >
> > To address your primary concern regarding the scalability of our method to higher-dimensional motion planning tasks, we have added a new experiment involving a 4-DOF robotic manipulator. These results are now presented in Figures 4 and 8 of the updated manuscript (The main PDF contains the revisions, and the supplementary is the same document with the edits highlighted in red for convenience). This addition demonstrates the promise of our proposed Planning neural operator (PNO)-based method in scaling to higher degrees of freedom.
> >
> > During the discussion period, we’ve also made some additional improvements to strengthen the manuscript - which are summarized below for your reference. First, in the revised Table 2, we highlight the advantages of each component of the PNO architecture design in an ablation study format. In particular, we showcase (1) the importance of hard obstacle encoding for generalization across different environment geometries, and (2) the importance of the triangle inequality preservation, which significantly enhances generalization to varying goals. This discussion is expanded in Section 4. Secondly, we added two other sets of new experiments to highlight the flexibility of the proposed PNO architectures in handling physics-informed loss and its advantages in value function approximation (in comparison with PNT-Fields) detailed in the supplementary material (https://postimg.cc/xX0dmwM3) as well as Figure 3 respectively.
> >
> > We sincerely hope these updates address your concerns. If the new additions clarify and strengthen the work to your satisfaction, we would greatly appreciate it if you could let us know whether these changes could lead to an updated evaluation. If there are any remaining questions, we would be thankful and pleased to discuss them during the discussion period.
> >
> > Thank you once again for your time and invaluable feedback.

---

### Author Response · Authors · 2024-11-26
**Overall response**

We thank all reviewers for their helpful and insightful comments. We appreciate that the reviewers recognize the innovative use of operator learning as well as the theoretical foundation of our work. First, we address common questions and concerns of all reviewers and then we provide an individual response to each reviewer. We have attached two versions of the document. The main PDF contains the revisions, and the supplementary is the same document with the edits highlighted in $\color{red}\text{red}$ for convenience.

**Presentation of theoretical results**

We recognize that the mathematical formulation of the operator learning problem is challenging and have made the following edits to improve the presentation.
1. We added comments about the feasibility of Assumption 1 and 2, emphasizing that both assumptions are achievable in practice.
2. As suggested by Reviewer 6Xqc, we improved the diagram overviewing the architecture of our model and added explanations for the mathematical formulation of the layers in our model to Section 3. We hope that this clarifies the exact structure of PNO as well as solidifies the reasoning for each component.

**Ablation study and additional benchmark comparison**

We thank the reviewers for their suggestions to highlight the improvements in our architecture compared to the state of the art. We made the following additions.
1. We have altered the MovingAI city map section to highlight the advantages of
each component of our architecture in an ablation study format. In particular,
we showcase the importance of obstacle encoding in DAFNO and PNO over FNO
as well as the effect of the triangle inequality preservation in PNO as it achieves
much better generalization across different goals compared to DAFNO. These
results are presented in Table 2 with additional discussion in the exposition of
Section 4, ”Ablation Study: Moving AI City Maps”.
2. We added comparisons to P-NTFields [1] on the 3D iGibson environments. The idea behind P-NTFields is that during training the speed-field is slowly increased from $0.5$ to $\approx 1$ to eliminate local minima. As a result, as shown in Figure 3, P-NTFields indeed generates very nice paths. However, quantitatively, due to this training methodology, the value function captured by P-NTFields differs in magnitude from FMM because the FMM input velocity-field is a constant $1$. Despite this difference in magnitude, one can see that in the contour plots, FMM and P-NTFields largely match. We added a sentence in the paper to highlight this clarification.

**Scalability and application to manipulators**

To showcase the ability of our approach to handle more complex tasks, we introduced a new experiment applying PNO to a 4-DOF robot manipulator planning problem. In the 4-DOF problem, we demonstrate that PNO is able to compute value functions in $C-$space (Configuration space). We show visualizations of the value function predictions in Figure 8 in the Appendix and two examples of manipulator path executions in Figure 4.

In future work, we plan to apply the PNO architecture to real-world robot manipulators with higher DOF but we emphasize that our focus in this work is on the foundation of the operator learning approach and its strengths. To extend our method to higher DOF manipulators, we recognize the challenge of using point-wise training data and thus will need to invoke different representations of the environment. However, this will introduce new challenges of ensuring obstacle avoidance and the satisfaction of the triangle inequality in latent space, leaving an exciting direction for future research.

**Overall impact and contributions:**

We briefly summarize the impact and contributions of the paper. This paper lays the groundwork for developing super-resolution value function approximation in continuous function space by introducing the first connection between neural operators and the Eikonal PDE in motion planning problems. We prove the existence of an arbitrarily good neural-operator approximation theoretically and highlight the architectural advantages of super-resolution operator-based motion planning across 2D, 3D, and 4D experiments. We then give an example of the utility of super-resolution value function approximation by employing the PNO value predictions as heuristics for motion planning algorithms, proving both $\epsilon-$consistency of the heuristic as well as showing improvement in the number of expanded nodes in $A^\ast$. The key advantage of the approach across all experiments is PNO's ability to *generalize* to new environments, while enabling super-fine discretization for accurate planning. Thus, this work opens a new perspective for designing neural motion planners that function across different environments while maintaining the high accuracy needed for motion planning.

[1] R. Ni and A. H. Qureshi. Progressive learning for physics-informed neural motion planning. Robotics: Science and Systems, July 2023.

---

### Meta-Review · Area_Chair_nfA9 · 2024-12-25

**Metareview:**

The paper introduces the Planning Neural Operator (PNO), a model that predicts value functions for motion planning by learning an operator from cost function space to value function space, demonstrating zero-shot super-resolution, accurate predictions, and significant efficiency improvements in motion planning across 2D, 3D, and robotic manipulator tasks.

The reviewers highlighted several strengths of the paper, including (1) its novelty, (2) applicability across multiple tasks, (3) advantages over existing baselines (VIN, NTFields, and P-NTFields), and (4) its clear and well-structured presentation. However, they also raised concerns about the simplicity of the experimental setups and questioned the method's scalability to larger-scale, higher-dimensional spaces and real-world scenarios.

During the Author-Reviewer Discussion phase, extensive discussions occurred between the authors and Reviewer 6Xqc. The authors provided additional experimental results and clarifications that clearly strengthened the paper. While incorporating more challenging experiments, particularly in real-world settings, would further enhance the paper's impact, the AC acknowledges the theoretical contribution and believes it represents a valuable addition to the ICLR community.

The AC recommends acceptance but encourages the authors to carefully address both pre- and post-rebuttal comments to address any remaining concerns in future revisions.

**Additional Comments On Reviewer Discussion:**

During the Reviewer Discussion phase, Reviewer C6NW remained negative but failed to provide compelling arguments against the paper. After thoroughly reviewing the reviews, rebuttal, and discussion, the AC concludes that the authors have sufficiently addressed the key concerns and recommends accepting the paper.

---

### Decision · Program_Chairs · 2025-01-22

Accept (Poster)